# Panoptic Captioning: An Equivalence Bridge for Image and Text

**Kun-Yu Lin**   **Hongjun Wang**   **Weining Ren**   **Kai Han**\*

Visual AI Lab, The University of Hong Kong

kunyulin@hku.hk   {hjwang,weining}@connect.hku.hk   kaihanx@hku.hk

## Abstract

This work introduces *panoptic captioning*, a novel task striving to seek the minimum text equivalent of images, which has broad potential applications. We take the first step towards panoptic captioning by formulating it as a task of generating a comprehensive textual description for an image, which encapsulates all entities, their respective locations and attributes, relationships among entities, as well as global image state. Through an extensive evaluation, our work reveals that state-of-the-art Multi-modal Large Language Models (MLLMs) have limited performance in solving panoptic captioning. To address this, we propose an effective data engine named *PancapEngine* to produce high-quality data and a novel method named *PancapChain* to improve panoptic captioning. Specifically, our *PancapEngine* first detects diverse categories of entities in images by an elaborate detection suite, and then generates required panoptic captions using entity-aware prompts. Additionally, our *PancapChain* explicitly decouples the challenging panoptic captioning task into multiple stages and generates panoptic captions step by step. More importantly, we contribute a comprehensive metric named *PancapScore* and a human-curated test set for reliable model evaluation. Experiments show that our PancapChain-13B model can beat state-of-the-art open-source MLLMs like InternVL-2.5-78B and even surpass proprietary models like GPT-4o and Gemini-2.0-Pro, demonstrating the effectiveness of our data engine and method. Project page: https://visual-ai.github.io/pancap/

## 1  Introduction

Representing images by textual descriptions is a fundamental topic in computer vision and natural language processing fields [1, 2], which benifits various applications, *e.g.*, cross-modal retrieval [3, 4], multi-modal learning [5, 6, 7, 8, 9, 10], safe content generation [11, 12]. While prior works have explored various image caption formats, identifying the most effective format remains an open challenge. The most concise captions, which describe only primary entity categories, often sacrifice critical details like entity attributes. Conversely, highly detailed representations, such as paragraphs detailing all pixel-level semantics and their interrelations, are computationally burdensome due to their length. Inspired by these considerations, this work conceives of finding the *minimum text equivalent* of an image, an ambitious yet challenging goal, which aims to develop a concise textual description that comprehensively captures its *essential* semantic elements. Conceptually, achieving minimal text equivalence for images can be seen as aligning images and text in the *data* space, while existing image-text alignment models like CLIP [13] perform this in the *embedding* space. Such text representations would maximize the utility of image information for learning and downstream applications.

This work introduces the task of *panoptic captioning*, which strives to seek the minimum text equivalent of images. Our work serves as the initial effort towards this challenging task. To make the

---

\*Corresponding Author

39th Conference on Neural Information Processing Systems (NeurIPS 2025).

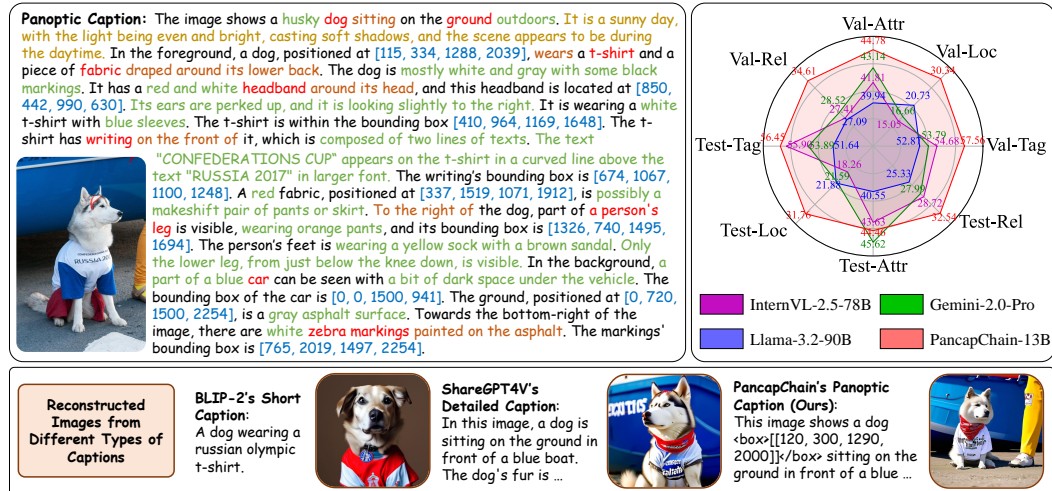

Figure 1: Top Left: An example to demonstrate our proposed *panoptic captioning* task, which is formulated as generating a comprehensive textual description encapsulating all entities, their respective locations and attributes, relationships among entities, as well as global image state for a given image. Top Right: We report several models' performance on four distinct dimensions on the validation and test sets of our SA-Pancap benchmark. The figure shows that three state-of-the-art Multi-modal Large Language Models (MLLMs) struggle with panoptic captioning, while our proposed PancapChain performs generally better with a significantly smaller model size. Bottom: Image "reconstruction" by the text-to-image model PixArt-Σ [14] with different types of captions. Best viewed in color.

problem tractable, we formulate panoptic captioning as the task of generating a comprehensive textual description for an image, which captures all entity instances, their respective locations and attributes, relationships among entity instances, as well as global image state (see Figure 1 (top left)). This formulation serves as a reasonable approximation to our conceptual "minimum text equivalence", which captures basic semantic elements (*e.g.*, center dog) for comprehensiveness while excluding less critical or subtle details (*e.g.*, tiny particles on the ground) for conciseness. In contrast to prevailing captioning works that vaguely specify locations by pure words [15, 1, 2, 4, 16, 17, 5], *e.g.*, BLIP-2's brief captions and ShareGPT4V's detailed captions, our panoptic captioning stands out for its text comprehensiveness. By accurately localizing entity instances using bounding boxes, it offers an effective and efficient way to describe position and occupied region of an entity instance with only several coordinate numbers. Figure 1 (bottom) shows our panoptic captioner performs better image "reconstruction" from captions due to better comprehensiveness.

Due to the fundamental formulation differences from existing captioning works, their metrics cannot effectively evaluate model performance in panoptic captioning. To this end, we propose a new metric named PancapScore, which groups semantic content into distinct dimensions based on task formulation and then conducts evaluation on each dimension. PancapScore first evaluates semantic tagging and instance localization by entity instance matching, and then assesses attribute, relation and global state in a flexible question-answering manner. Our experiments demonstrate that PancapScore aligns closely with human judgement. Based on PancapScore, we conduct a thorough evaluation on state-of-the-art Multi-modal Large Language Models (MLLMs). As shown in Figure 1 (top right), existing MLLMs have limited performance in panoptic captioning, revealing its challenging nature.

To address this new and challenging task, we propose an effective data engine named PancapEngine to produce high-quality data in a detect-then-caption manner. Specifically, we first detect diverse categories of entities in images, by associating class-agnostic detection with image tagging. We then employ state-of-the-art MLLMs to generate comprehensive panoptic captions using entity-aware prompts, while ensuring data quality by caption consistency across MLLMs. Based on PancapEngine, we contribute a new SA-Pancap benchmark composed of high-quality auto-generated data for training and validation, and additionally provide a human-curated test set for reliable evaluation.

Furthermore, we propose a novel method named PancapChain to improve panoptic captioning. The key idea is to decouple the challenging panoptic captioning task into multiple stages and train the model to generate panoptic captions step by step. PancapChain first localizes entity instances, then assigns semantic tags to instances, and finally generates panoptic captions. Surprisingly, our

PancapChain-13B model beats state-of-the-art large open-source MLLMs like InternVL-2.5-78B, and even surpasses proprietary models like GPT-4o and Gemini-2.0-Pro. Additionally, our model can facilitate downstream image-text retrieval tasks by transforming images into captions and performing text retrieval. Table 1 shows that our model achieves superior performance on the challenging DOCCI dataset, which outperforms a state-of-the-art image-text alignment model ALIGN without specialized training data and module designs, and beats the state-of-the-art detailed captioner ShareGPT4V. These experiments in panoptic captioning and downstream image-text retrieval demonstrates the effectiveness and application value of our task and model.

Table 1: Image-text retrieval results on DOCCI [4] in Recall@1.

| Models | Model Type | R@1 |
|---|---|---|
| ALIGN [18] | Image-Text | 59.9 |
| BLIP [15] | Text-Text | 47.3 |
| ShareGPT4V [5] | Text-Text | 59.6 |
| PancapChain | Text-Text | **61.9** |

Our contributions are summarized as follows: First, we introduce the novel panoptic captioning task, which strives to seek the minimum text equivalent of an image—an ambitious yet challenging goal. We formulate it as the task of generating a comprehensive textual description composed of five distinct dimensions, and contribute a comprehensive PancapScore metric for reliable evaluation. Second, we propose an effective data engine named PancapEngine to produce high-quality data. We also contribute the SA-Pancap benchmark for model training and evaluation, which includes a high-quality validation set and a human-curated test set for reliable evaluation. Third, we propose a simple yet effective method named PancapChain to improve panoptic captioning, which decouples the challenging panoptic captioning task into multiple subtasks. Extensive experiments demonstrate the effectiveness and value of our task and model.

## 2    Related Work

**Image Captioning.** Image captioning is a fundamental topic in computer vision and nature language processing fields [1, 2]. It aims to describe the visual content of an image using meaningful and syntactically correct sentences. In early years, many methods have been proposed to address image captioning, mainly based on RNNs and Transformers [19, 20, 21, 22, 23, 24, 25, 26, 27, 28]. Traditional image captioning adopts evaluation metrics like BLEU [29], METEOR [30] and CIDEr [31], which leverage N-gram and lexical similarity with human-annotated captions. Above early works focus on short captions lacking details. Additionally, scene-graph-based captioning methods [32] first generate scene graphs and then produce captions. However, these methods are restricted to fixed, predefined categories of objects and relationships, and their performance is limited by the subsequent caption integration process and the need for additional models. Recently, owing to the development of vision-language pre-training, Multi-modal Large Language Models (MLLMs) have been equipped with the capabilities of generating detailed image captions [15, 33, 34, 35, 17]. These detailed captioning works also promote the development of MLLMs, *e.g.*, using detailed captions in pre-training boosts performance on downstream tasks [5, 6]. Due to the unstructured nature and rich semantic complexity of detailed captions, evaluating model performance in detailed captioning presents significant challenges. Recent works have developed various types of metrics to evaluate long captions [36, 37, 38, 39, 40, 41, 42, 43, 44].

Previous captioning works vaguely specify entity locations by pure words. Unlike these works, our panoptic captioning introduces a more comprehensive and economic caption format, which accurately localizes entity instances using bounding box coordinates. This enables better descriptions for the positions and occupied regions of entity instances, and helps distinguishing instances with similar attributes. Accordingly, we propose a new PancapScore metric for comprehensive evaluation. Another related task is dense captioning [22, 45], which generates short captions for individual regions in an image. Typically, this task focuses on a limited set of entity categories and does not account for the relationships between entities. Some recent Grounded MLLM works have explored the joint task of image captioning and visual grounding [46, 47, 48]. However, they rely on additional specialized modules to achieve localization capabilities, and they usually generate brief captions lacking details for a whole image (*e.g.*, GLaMM [46]). In contrast, our work integrates localization capabilities into detailed captioning through pure textual descriptions using a unified LLM, taking an initial step to explore the minimum text equivalent of images.

**Vision-Language Models.** Recently, significant advancements have been made in Vision-Language Models (VLMs). A typical type of VLMs is based on image-text matching [13, 18, 49, 50, 51, 52, 53, 54], which achieves remarkable zero-shot image understanding performance and initiates the era of

open-world understanding [55, 56, 57, 58, 59, 60, 61, 62, 63, 64, 65, 66, 67, 68, 69]. Another type of VLMs is called Multi-modal Large Language Models (MLLMs) [34, 35, 70, 71, 72, 73, 74, 75, 70, 76], which is motivated by the remarkable success of Large Language Models (LLMs) [77, 78, 79, 80] in instruction following [81], in-context learning [82] and reasoning [83]. MLLMs propose to integrate the power of LLMs with multi-modal perception and comprehension, which enables multifunctional visual understanding and shows remarkable performance in various tasks. In addition, recent Grounded MLLMs explore integrating location-aware understanding abilities into MLLMs, enabling region-aware downstream tasks [84, 85, 86, 87, 46, 88, 89, 90, 91, 47, 48]. Although significant efforts have been devoted to advance MLLMs, our work reveals that state-of-the-art MLLMs exhibit limited performance in panoptic captioning, which is fundamental for image and multi-modal understanding.

## 3 Panoptic Captioning

### 3.1 Task Definition

In this work, we formulate panoptic captioning as the task of generating a comprehensive textual description for an given image, which encapsulates all entity instances, their respective locations and attributes, relationships among instances, as well as global image state. Specifically, we group the semantic content in panoptic captions into five dimensions, which are detailed as follows:

***Semantic tag*** refers to the category label assigned to each entity instance in an image. Panoptic captioning requires identifying all entity instances and assigning category label to each instance. Following Kirillov et al. [92], we define "entities" as both countable objects (things) such as people and animals, and amorphous regions (stuff) such as grass, sky, and road.

***Location*** refers to the spatial positions of entity instances, which are represented in terms of bounding boxes. Specifically, following previous detection works [93], the bounding box of an entity instance is denoted by $(x_1, y_1, x_2, y_2)$, where $(x_1, y_1)$ and $(x_2, y_2)$ denote the top-left and bottom-right corners of the box, respectively. By introducing bounding boxes, panoptic captions can more accurately describe the locations and occupied regions of entity instances, which also helps distinguishing entity instances with similar attributes more easily.

***Attribute*** refers to characteristics or properties that describe an entity instance's appearance, state or quality. The attribute dimension encompasses a wide range of semantic content types, *e.g.*, color, shape, material, texture, type, text rendering. Describing attributes can provide a detailed understanding for an entity instance, enabling accurate entity identification and image analysis [94, 41, 43].

***Relation*** refers to connections or interactions between different entity instances within an image. The relation dimension encompasses a wide range of semantic content types, such as position relation (*e.g.*, A is behind B), part-whole relation (*e.g.*, A is a part of B) and action relation (*e.g.*, A kicks B). Describing the relations between entities is crucial for understanding the image's structure, context, and dynamics [94, 41, 43].

***Global image state*** refers to the overall characteristics of an image that provide a holistic understanding of its content, without focusing on specific entity instances within the image [94]. For example, global image state includes lighting conditions and color palette.

Figure 1 (top left) demonstrates an example for the task definition. Overall, by considering the above five distinct dimensions, panoptic captioning leads to a comprehensive textual representation by capturing all basic semantic elements. It should be noted that our proposed panoptic captioning differs from conventional captioning works [15, 5, 4, 16, 17] mainly in text comprehensiveness. Instead of vaguely specifying entity locations by pure words, panoptic captioning requires models to accurately localize entity instances using bounding boxes, which enables better descriptions for positions and occupied regions of instances and helps distinguishing instances with similar attributes more easily.

### 3.2 The PancapScore Metric

Prior works in detailed captioning [43, 41] have demonstrated that evaluating the quality of detailed captions presents significant challenges due to their free-form nature and rich content. Evaluating models in panoptic captioning becomes even more complex, as it requires an additional assessment of bounding box accuracy. To address this challenge, we propose a new metric named PancapScore for comprehensive and reliable evaluation. PancapScore systematically categorizes the content in a panoptic caption into five distinct dimensions and evaluate model performance on each dimension separately, faithfully aligning with the task objective.

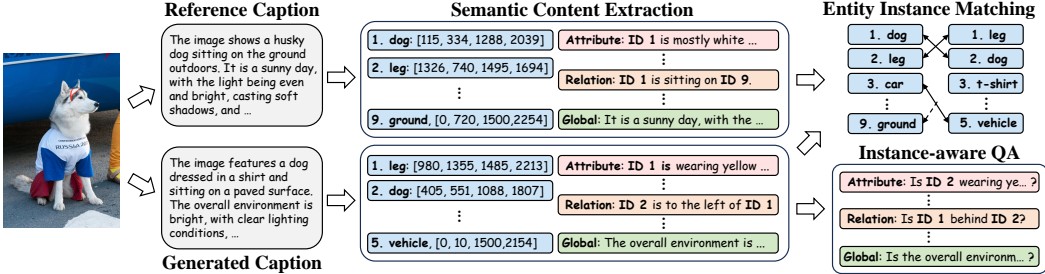

Figure 2: An overview of our proposed PancapScore metric. PancapScore first extracts semantic content from captions, and then evaluates model performance by entity instance matching and instance-aware question answering (QA).

Specifically, given an image, PancapScore evaluates the quality of a generated panoptic caption using the ground-truth caption as the reference. PancapScore first extracts all semantic content from captions and groups them into five dimensions. Based on extracted semantic content, PancapScore evaluates semantic tagging and instance localization based on entity instance matching. Then, PancapScore evaluates attribute, relation and global state in a question-answer manner, and finally obtains the overall score considering all five dimensions. An overview of our proposed PancapScore metric is given in Figure 2.

**Semantic Content Extraction.** First of all, we propose to extract all the semantic content from a generated caption and group these content into five dimensions, namely semantic tag, location, attribute, relation and global state. The content extraction is implemented using state-of-the-art Large Language Models (LLMs), as they have strong understanding capabilities on textual descriptions and instructions. To facilitate the understanding of LLMs, we carefully design prompts with in-context text examples included.

Figure 2 includes an example of the extracted semantic content. Specifically, the extracted semantic content is organized item by item, and each item is a basic unit of semantic information. The leading items specify the semantic tags and locations of entity instances. Each entity instance is associated with a unique semantic-tag item with a unique instance ID. The following items describe attribute and relation information. Each of these items is associated with an instance ID, indicating that it describes the attribute or relation of the corresponding instance. The last items describe the global state, and these items are not tied to specific instances. During evaluation, we will separately extract semantic content from the generated and reference captions.

**Entity Instance Matching.** After extracting semantic content, we conduct an instance matching between the generated and reference captions to evaluate semantic tagging and instance localization. Formally, the ground-truth and predicted entity instance sets are denoted by $\{(t_i, b_i)\}_{i=1}^n$ and $\{(\hat{t}_j, \hat{b}_j)\}_{j=1}^m$. $(t_i, b_i)/(\hat{t}_i, \hat{b}_i)$ denote the semantic tag and bounding box of the $i$-th ground-truth/predicted instance. $n$ and $m$ denote the total number of instances. Accordingly, the entity instance matching is formulated as an optimal one-to-one matching problem as follows:

$$O = \arg\max_{O_{i,j}} \sum_{i=1}^n \sum_{j=1}^m (\mu \cdot s_{i,j} + \text{iou}(b_i, \hat{b}_j)) \cdot O_{i,j},$$

where $O$ is an assignment matrix and $O_{i,j} \in \{0, 1\}$ indicates if $(t_i, b_i)$ and $(\hat{t}_j, \hat{b}_j)$ are matched (i.e., $O_{i,j} = 1$ indicates that the $i$-th ground-truth instance matches the $j$-th predicted instance). $s_{i,j}$ denotes the similarity between $t_i$ and $\hat{t}_j$, which considers synonyms and embedding similarity. $\text{iou}(b_i, \hat{b}_j)$ denotes the box IoU, and $\mu = 10$ is a weight coefficient to raise the priority of tagging.

Based on the entity instance matching, we measure the model performance in semantic tagging and instance localization. First, we consider two matched instances to be semantically consistent if their tags share synonymous nouns or have similar text embeddings, i.e., $s_{i,j} \geq \delta_t$. By considering all entity instances, we compute precision and recall in semantic tagging. Precision measures the proportion of correctly predicted tags among all predictions, while recall measures the proportion of correctly identified ground-truth instances. Based on precision and recall, we compute the F-score for an overall measurement for semantic tagging. Besides, we consider two semantic-consistent instances to be location-consistent if $\text{iou}(b_i, \hat{b}_j) \geq \delta_l$. Following a similar way in semantic tagging, we obtain F-score for instance localization by localization consistency. Please see Appendix for more details.

**Instance-aware Question Answering.** Based on entity instance matching, we evaluate models' performance in attribute, relation and global state in a flexible question-answering manner. First, we generate questions based on the semantic content of the generated caption, where each question verifies whether an attribute/relation/global state item is described in the reference caption. For each attribute/relation item, the associated instance ID in the generated caption is mapped to their corresponding instance ID in the reference caption. For example, if the generated caption contains an item "ID 2 is red" and the "ID 2" corresponds to the "ID 3" in the reference caption, we generate a question like "Is ID 3 red?". To enable robust evaluation, we generate a pair of questions for each semantic item: one with a "Yes" answer and the other with a "No" answer.

We employ a state-of-the-art LLM to answer questions according to the reference caption. In this process, we instruct the employed LLM with a carefully designed prompt, where in-context text examples are included. If the LLM outputs the same answers as the preset ones, we consider the corresponding semantic item to be a correct prediction. We measure precision in attribute/relation/global state by computing how many questions are correctly answered. Additionally, we measure recall by generating questions from the reference caption and evaluate the LLM's answers based on the generated caption. Based on precision and recall, we obtain F-scores for an overall measurement in attribute, relation and global state.

Overall, our PancapScore metric includes five F1-scores from dimensions of tagging, localization, attribute, relation and global state. The five scores are denoted by $s_t$, $s_l$, $s_a$, $s_r$ and $s_g$, respectively. Based on these scores, the overall score of our metric is formulated as: $s = s_t + s_l + s_a + s_r + \lambda_g s_g$. Since an image usually has much fewer global state items compared with other dimensions (*i.e.*, usually one or two items), we introduce the coefficient $\lambda_g = 0.1$ to downweight the global state's score. Experiments in Appendix show that PancapScore aligns closely with human judgement.

## 4 Data Engine and Benchmark

### 4.1 PancapEngine

To address this new and challenging task, our work proposes an effective automated data engine named PancapEngine to produce high-quality data. Our PancapEngine first detects diverse categories of entities in images using an elaborate entity detection suite. We then employ state-of-the-art MLLMs to generate comprehensive panoptic captions using entity-aware prompts, ensuring the data quality by caption consistency across different MLLMs.

**Entity Detection Suite.** Existing detection or segmentation datasets are often limited to a small number of predefined categories[1], thus directly using these datasets will limit the entity diversity of panoptic captioning data. To improve entity diversity, we propose to associate class-agnostic detection with image tagging for detecting diverse categories of entities in a given image. Specifically, we first detect entity instances using a state-of-the-art class-agnostic detector OLN [95], and the resulting set of regions is denoted by $\mathcal{R}$. We then assign semantic tags to regions by a state-of-the-art image tagging model RAM [96]. For each region in $\mathcal{R}$, we crop the region from the image and feed it into RAM to obtain its semantic tag. RAM can recognize 6,400+ common entity categories, which is much more diverse than existing detection datasets. In addition, we integrate two specialized class-aware detectors, namely Grounding-DINO [99] and OW-DETR [100], to identify instances missed by OLN. We aggregate all entity categories from OLN's detected regions and utilize this aggregated category set as input prompts for Grounding-DINO and OW-DETR to enable class-aware detection. The resulting region set from class-aware detectors is denoted by $\mathcal{R}'$. We then merge the two sets $\mathcal{R}$ and $\mathcal{R}'$, and remove redundant regions based on IoU. In this case, we will add a region from $\mathcal{R}'$ to $\mathcal{R}$, if this region has low IoUs with all the regions in $\mathcal{R}$. We do not use non-maximum suppression to remove redundant proposals as different detectors produce confidence scores in varying ranges. With a comprehensive set of detected entity instances, we proceed to produce detailed panoptic captions.

**Entity-aware Caption Generation.** Based on detected entity instances in images, we construct entity-aware prompts and instruct state-of-the-art MLLMs to generate panoptic captions. An entity-aware prompt includes all detected entity instances in the query image, associated with semantic tags and locations. In addition, the prompt explicitly specifies attribute, relation and global state types to help MLLMs discover more semantic content. In-context examples are also included in the prompt for better instruction following. We employ Gemini-Exp-1121 and Qwen2-VL-72B

---

[1]For example, COCO [93, 97] includes 80 objects and 91 stuff, and Object365 [98] includes 365 objects.

to generate required captions due to their strong image understanding and instruction-following capabilities. Specifically, we first employ Gemini-Exp-1121 to generate required captions using entity-aware prompts. Similarly, we employ Qwen2-VL-72B to generate required captions and conduct quality verification by caption consistency. We drop a caption generated by Gemini-Exp-1121, if it has low consistency with the corresponding Qwen-generated caption. We adopt our PancapScore metric to measure caption consistency, without considering the location dimension. This is because we empirically find that Qwen2-VL-72B has much weaker localization capabilities than Gemini-Exp-1121 when entity-aware prompts are given. Overall, by entity-aware generation and quality verification, our PancapEngine can generate high-quality panoptic captions.

## 4.2    The Proposed SA-Pancap Benchmark

Based on our PancapEngine, we contribute a new SA-Pancap benchmark for the panoptic captioning task. We select SA-1B [101] as the data source due to its high image quality and data diversity. Overall, our SA-Pancap benchmark consists of 9,000 training and 500 validation images paired with auto-generated panoptic captions, and 130 test images paired with human-curated panoptic captions.

Our validation and test sets consist of diverse images, paired with high-quality panoptic captions. Specifically, based on PancapScore, we select the images paired with the most high-quality panoptic captions to construct the validation set. Additionally, we ask human labors to refine model-generated panoptic captions of the test set, following a rigorous curation process. To ensure the data diversity of the validation and test sets, we leverage DINOv2 [102] to select distinctive images. Specifically, we enforce a DINOv2 feature similarity threshold of 0.2 between any two images within these sets, thereby preserving a high degree of visual variability. Additionally, we enforce a DINOv2 feature similarity threshold of 0.5 between training images and validation/test images.

In summary, SA-Pancap consists of high-quality panoptic captions, which comprehensively covers diverse categories of entities and rich semantic content in location, attribute, relation and global state dimensions. Table 2 shows a comparison between our SA-Pancap with previous captioning benchmarks. Compared with detailed captioning

Table 2: Compared with previous benchmarks, our SA-Pancap provides more comprehensive captions, associated with diverse entity instances and accurate location annotations.

| Benchmarks | Location | Instance | Category | Sample | Token |
|---|---|---|---|---|---|
| DCI [103] | ✗ | - | - | 7.8K | 148.0 |
| DOCCI [4] | ✗ | - | - | 14.6K | 135.7 |
| IIW [16] | ✗ | - | - | 9.0K | 217.2 |
| SG4V [5] | ✗ | - | - | 1.2M | 192.0 |
| DenFu [6] | ✗ | - | - | 1.0M | 254.7 |
| GCG [46] | ✓ | 2.9 | 1329 | 56.9K | 27.2 |
| SA-Pancap | ✓ | **6.9** | **2429** | 9.6K | **345.5** |

benchmarks, our SA-Pancap has more detailed captions with more tokens and associates entity instances in captions with location annotations. Compared with GCG [46] involving short captions, SA-Pancap provides much more comprehensive captions with diverse categories of entities and more instances per image.

## 5    PancapChain

To improve panoptic captioning, we propose a novel method named PancapChain. Our key idea is to decouple the challenging panoptic captioning task into multiple stages and train the model to generate panoptic captions step by step, as an image contains rich semantic elements. This is inspired by cognitive science works [104, 105], which has demonstrated that humans struggle to perceive and comprehend all elements within an complex scene simultaneously. Accordingly, given an image $\mathbf{Q}^v$, our PancapChain proposes to generate a panoptic caption $\hat{\mathbf{A}}$ by four stages, namely, entity instance localization, semantic tag assignment, extra instance discovery, panoptic caption generation, denoted as $\mathbf{S}_{\{\text{Loc, Tag, Disc, Cap}\}}$. We show an overview of PancapChain in Figure 3 and detail it as follows.

**Entity Instance Localization ($\mathbf{S}_{\text{Loc}}$).** In the first stage, we propose to localize entity instances, as entity instances are the foundation for describing images. To this end, for the image $\mathbf{Q}^v$, we extract the bounding boxes of instances from the ground-truth caption $\mathbf{A}$, and construct an image-text pair $\{\mathbf{Q}^v, \mathbf{A}^L\}$ for training. $\mathbf{A}^L$ is the localization text consisting of bounding boxes of all instances, which are concatenated by commas. Following ASMv2 [85], each bounding box is represented by the text in the format of "<box>[[$x_1, y_1, x_2, y_2$]]</box>", where $(x_1, y_1)$ and $(x_2, y_2)$ denote the top-left and bottom-right corners of the box. All bounding boxes are normalized to integer values within $[0, 1000)$, and images are resized to a fixed size.

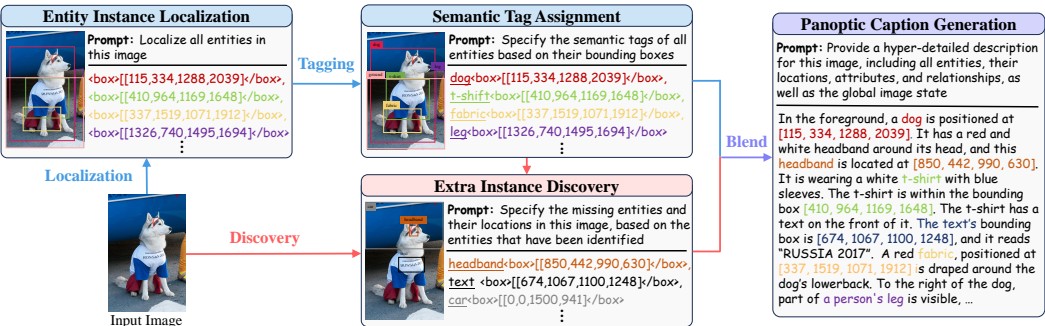

Figure 3: An overview of our proposed PancapChain method. PancapChain explicitly decouples the challenging panoptic captioning task into four stages, namely entity instance localization, semantic tag assignment, extra instance discovery and panoptic caption generation.

**Semantic Tag Assignment ($S_{\text{Tag}}$).** In the second stage, based on the localization text, we propose to assign semantic tags to the localized entity instances. To this end, we extract the semantic tags of instances from the ground-truth caption, with each associated with one bounding box, and then construct an image-text tuple $\{\mathbf{Q}^v, \mathbf{A}^L, \mathbf{A}^I\}$ for training. $\mathbf{A}^I$ is the instance text consisting of semantic tags and bounding boxes of all instances, which are concatenated by commas. Specifically, each entity instance is represented by the text in the format of "`tag <box>[[`$x_1, y_1, x_2, y_2$`]]</box>`", where "`tag`" denotes the ground-truth semantic tag. For example, "`dog <box>[[100, 200, 500, 600]]</box>`" means there is a `dog` located within the box `[100, 200, 500, 600]`. In this stage, $\mathbf{A}^L$ is introduced in the textual prompt to instruct model training. During inference, the predicted location text $\hat{\mathbf{A}}^L$ is included in the prompt.

**Extra Instance Discovery ($S_{\text{Disc}}$).** Since an image contains numerous entity instances, it is not trivial to identify all instances at once. Therefore, we introduce an additional stage to detect instances that are missed in the first two stages. Specifically, for the image $\mathbf{Q}^v$, we construct an image-text tuple $\{\mathbf{Q}^v, \mathbf{A}^I_1, \mathbf{A}^I_2\}$ for training, where $\mathbf{A}^I_1$ is included in the textual prompt and $\mathbf{A}^I_2$ is used as the ground-truth for supervision. We randomly split the ground-truth instance set into two parts, and construct two instance text namely $\mathbf{A}^I_1$ and $\mathbf{A}^I_2$. The text $\mathbf{A}^I_1$ contains boxes and tags of those instances that have been discovered, while $\mathbf{A}^I_2$ indicates the other instances to be discovered. In our design, $\mathbf{A}^I_2$ contains fewer instances than $\mathbf{A}^I_1$ to simulate the discovery of previously missed instances. During inference, we include the predicted instance text $\hat{\mathbf{A}}^I$ in the prompt and then predict the extra instance text $\hat{\mathbf{A}}^E$.

**Panoptic Caption Generation ($S_{\text{Cap}}$).** Finally in the fourth stage, we generate panoptic captions based on identified entity instances in early stages. Formally, we construct an image-text tuple $\{\mathbf{Q}^v, \mathbf{A}^I, \mathbf{A}\}$ for training, where $\mathbf{A}$ is the ground-truth panoptic caption of the image. The bounding boxes in $\mathbf{A}$ will be transformed into the same format as those in entity instance localization. The instance text $\mathbf{A}^I$ is included in the textual prompt for training. During inference, we should first aggregate the initial instance text $\hat{\mathbf{A}}^I$ and extra instance text $\hat{\mathbf{A}}^E$, and include the aggregated instance text in the prompt to predict caption $\hat{\mathbf{A}}$.

Overall, the training loss of our PancapChain is formulated as: $L(\hat{\mathbf{A}}^L, \mathbf{A}^L) + L(\hat{\mathbf{A}}^I, \mathbf{A}^I) + L(\hat{\mathbf{A}}^E, \mathbf{A}^I_2) + L(\hat{\mathbf{A}}, \mathbf{A})$, where $L(\cdot, \cdot)$ denotes the standard auto-regressive loss following LLaVA [34]. Different prompts are used in these four stages to instruct model training. During inference, our model generates captions stage by stage, according to the guidance of prompts.

## 6  Experiments

**Implementation Details.** Our model adopts the general LLaVA architecture [34], and it is initialized using the pre-trained ASMv2-13B [85] checkpoint due to its good grounding capabilities. We finetune our model on the training set of our SA-Pancap for two epochs using LoRA. For PancapScore, we use Qwen2.5-14B [106] as the LLM for semantic content extraction and question answering. The thresholds are set as $\delta_t = 0.5$ and $\delta_l = 0.5$. Please see Appendix for more details.

Table 3: Comparison with state-of-the-art MLLMs on the validation and test sets of SA-Pancap. Performance are measured by PancapScore, and we report the scores in tagging (Tag), location (Loc), attribute (Att), relation (Rel), global state (Glo), and the overall score (All). **Bold**/underlined indicates the best/second-best. We prioritize the overall score as the primary metric in model comparison.

| Models | Scale | Validation | | | | | | Test | | | | | |
|---|---|---|---|---|---|---|---|---|---|---|---|---|---|
| | | Tag | Loc | Att | Rel | Glo | All | Tag | Loc | Att | Rel | Glo | All |
| Molmo [108] | 72B | 52.06 | 10.03 | 36.88 | 25.90 | 76.78 | 132.53 | 50.92 | 14.00 | 38.10 | 19.96 | 68.49 | 130.55 |
| LLaVA-OV [109] | 72B | 54.20 | 13.79 | 38.94 | 27.80 | 85.52 | 143.28 | 53.62 | 15.16 | 41.52 | 25.63 | 82.39 | 144.17 |
| Qwen2-VL [107] | 72B | 49.85 | 12.92 | 37.83 | 24.71 | 86.30 | 133.96 | 48.19 | 12.90 | 38.48 | 20.44 | 84.13 | 128.42 |
| Qwen2.5-VL [86] | 72B | 54.08 | 19.70 | 40.00 | 27.24 | 85.34 | 149.54 | 54.42 | 25.11 | 42.33 | 26.32 | 87.12 | 156.89 |
| NVLM [110] | 72B | 54.69 | 10.78 | 42.49 | 30.40 | 86.21 | 146.97 | **57.79** | 11.53 | **46.48** | 29.48 | 78.60 | 153.14 |
| InternVL-2.5 [111] | 78B | 54.68 | 15.05 | 41.81 | 27.41 | 88.37 | 147.79 | 55.90 | 18.26 | 43.63 | 28.72 | 81.46 | 154.66 |
| Llama-3.2 [35] | 90B | 52.87 | 20.73 | 39.94 | 27.09 | 83.40 | 148.98 | 51.64 | 21.88 | 40.55 | 25.33 | 79.55 | 147.35 |
| GPT-4o [78] | Proprietary | 50.89 | 10.12 | 40.54 | 25.40 | **88.85** | 135.83 | 53.51 | 14.55 | 43.86 | 27.38 | 87.08 | 148.01 |
| Gemini-2.0-Pro [112] | Proprietary | 53.79 | 16.66 | 43.14 | 28.52 | 86.50 | 150.75 | 53.89 | 21.59 | 45.62 | 27.99 | **87.91** | 157.88 |
| LLaVA-1.5 [74] | 13B-Tuned | 54.92 | 27.76 | 41.27 | 28.69 | 81.94 | 161.84 | 54.33 | 30.57 | 41.81 | 30.62 | 75.73 | 164.92 |
| ShareGPT4V [5] | 13B-Tuned | 55.02 | 23.81 | 40.53 | 29.13 | 82.16 | 156.70 | 52.94 | 25.56 | 39.56 | 25.11 | 80.36 | 151.21 |
| PancapChain (Ours) | 13B-Tuned | **57.56** | **30.34** | **44.78** | **34.61** | 84.59 | **175.75** | 56.45 | **31.76** | 44.46 | **32.54** | 79.85 | **173.19** |

**Comparison with State-of-the-Arts.** Table 3 summarizes the comparison results between our proposed PancapChain and current state-of-the-art MLLMs on SA-Pancap. In this experiment, we carefully design prompts for these MLLMs to unleash their panoptic captioning power, where in-context examples are introduced for better instruct-following. From the table, we find that current MLLMs struggle with panoptic captioning, except the global state dimension. This is attributed to the challenging nature of the task, which requires models to comprehensively capture semantic content on five dimensions in images. We find that grounding capabilities are important for panoptic captioning, as Qwen2.5-VL [86] significantly outperforms Qwen2-VL [107]. By finetuning on our training set, our PancapChain-13B model significantly outperforms all state-of-the-art MLLMs in terms of the *overall* PancapScore metric. Also, except *the global state*, our PancapChain-13B usually achieves the best or second-best on each individual dimension. More importantly, our model is much smaller than state-of-the-art MLLMs, which demonstrates the effectiveness of our proposed data engine and model design. By directly finetuning on the training set, LLaVA-1.5 can also obtain good results, which demonstrates the high quality of our training data. Furthermore, our PancapChain still outperforms the tuned LLaVA-1.5 model, which demonstrates the effectiveness of our design.

**Ablation Study.** We use a one-stage model as our baseline, which is directly finetuned on the training set of SA-Pancap without specific designs. To improve panoptic captioning, we first introduce an extra instance discovery stage based on the baseline. This model variant first predicts semantic tags and locations of instances for an input image and then generates panoptic captions based on predicted tags and locations. Table 4 shows introducing this instance discovery stage improves the performance on attribute and relation dimensions, since we decouple the challenging task into two relatively easier subtasks. Next, we decouple the instance discovery stage into entity instance localization and semantic tag assignment stages, as introduced in Sec. 5. As shown by "w/ $S_{\{Loc, Tag\}}$", this variant obtains notable improvement in semantic tagging. As shown by "Full (w/ $S_{\{Loc, Tag, Disc\}}$)", by further introducing the proposed extra instance discovery stage, our model obtains notable overall improvement.

Table 4: Ablation study on the validation set of SA-Pancap. $S_{Loc, Tag, Disc}$ refer to our entity instance localization, semantic tag assignment and extra instance discovery stages, respectively. The caption generation stage $S_{Cap}$ is required in all cases.

| Models | Tagging | Location | Attribute | Relation |
|---|---|---|---|---|
| Baseline | 56.38 | 29.30 | 43.06 | 31.64 |
| w/ Instance Discovery | 56.47 | 29.61 | 43.71 | 32.62 |
| w/ $S_{\{Loc, Tag\}}$ | 57.04 | 29.83 | 43.76 | 33.69 |
| Full (w/ $S_{\{Loc, Tag, Disc\}}$) | 57.56 | 30.34 | 44.78 | 34.61 |

This is because that our model accurately identifies more entity instances and accordingly boosts the prediction on attribute and relation dimensions. In summary, our full model improves the baseline by 6.5+% in terms of the overall score.

**Image Reconstruction.** To qualitatively demonstrate the effectiveness of our model, we conduct an image reconstruction experiment by associating captioners with text-to-image generation models.

| Original Image | PancapChain | Qwen2.5-VL | ShareGPT4V | BLIP-2 |
| --- | --- | --- | --- | --- |

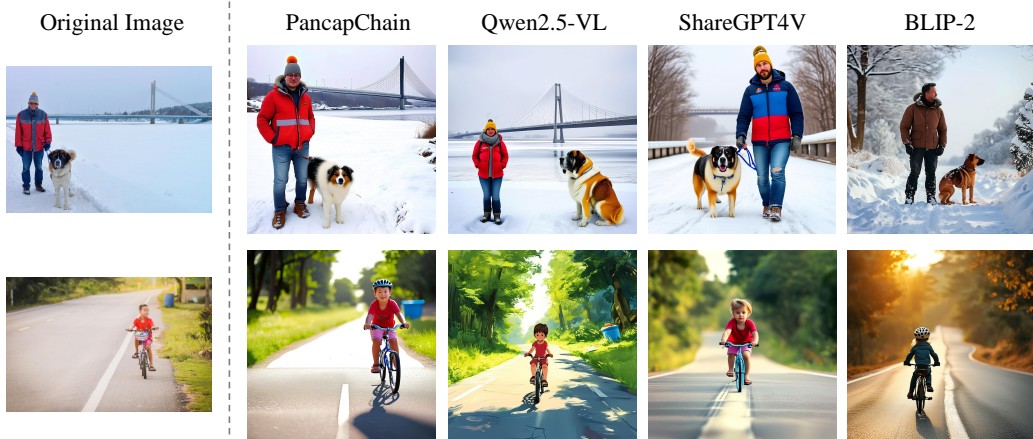

Figure 4: Image reconstruction using PixArt-$\Sigma$ [14]. Compared with baseline models, our PancapChain can better capture image details, and thus lead to better image reconstruction.

This experiment serves as a proxy for evaluating the completeness of image descriptions, *i.e.*, if a caption captures all essential visual elements, a text-to-image model would be able to reconstruct an image similar to the original one. Specifically, we generate a caption for an input image using a captioner, and then adopt a text-to-image generation model PixArt-$\Sigma$ [14] to generate a new image. As shown in Figure 4, PixArt-$\Sigma$ associated with our PancapChain model can generate more similar images to the original images than baseline models, which demonstrates the effectiveness of our model in comprehensively capturing image details. Please refer to the Appendix for more implementation details, results and discussions.

For more experiment results, *e.g.*, image-text retrieval and ablation study, please refer to the **Appendix**.

# 7 Conclusion and Discussion

This work introduces *panoptic captioning*, a novel task striving to seek the minimum text equivalent of an image—an ambitious yet challenging goal. We take the first step towards panoptic captioning by formulating it as a task of generating a comprehensive textual description composed of five dimensions. To study this task, we proposed a comprehensive metric for reliable evaluation, designed an effective data engine to produce high-quality data, and established a benchmark for model training and evaluation. To address this task, we proposed a decoupled learning method to generate captions step by step. Extensive experiments demonstrated the effectiveness and value of our task and model.

Our work reals that, despite remarkable progress in generating detailed textual descriptions, existing MLLMs still struggle with panoptic captioning, demonstrating their limitations in bridging image and text modalities. Also, our evaluation provides insights into the performance gap between open-source and proprietary models. By finetuning on high-quality data, our PancapChain-13B model beats state-of-the-art MLLMs, *e.g.*, InternVL-2.5-78B and Gemini-2.0-Pro, highlighting the potential of our task and model in bridging image and text modalities. Additionally, when paired with a text retriever, our model achieves superior performance in downstream image-text retrieval, demonstrating the practical utility of panoptic captioners. We believe that, developing more powerful panoptic captioners benifits various downstream applications or learning tasks, *e.g.*, multi-modal understanding.

Despite our efforts, panoptic captioning still faces numerous unresolved challenges, and our task formulation remains an approximation, not yet fully achieving our ultimate conceptual goal of "minimum text equivalence". Nevertheless, our work lays a solid foundation for future development, and will act as a catalyst to accelerate the progress of developing textual representations for images.

**Acknowledgments.** This work is supported by the Hong Kong Research Grants Council - General Research Fund (Grant No.: 17211024). The authors would like to thank Jiaming Zhou, Tianming Liang, Yi-Xing Peng, Yu-Ming Tang, An-Lan Wang, and Jonathan Roberts for their valuable suggestions. The authors also thank Samuel Albanie, Chengke Bu, Tianshuo Yan, Yuxian Li, Zhiwei Xia, Zhi Ouyang, Qiaochu Yang, Jialu Tang, Shiyang Chen, Hanyi Xiong, Shenru Zhang, and Boning Shao for their help and support for our project.

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

## Appendix

## A    Image-Text Retrieval Experiments

To demonstrate the application potential of our task and model, we apply our model to the downstream image-text retrieval task. In this experiment, we compare with two types of methods, namely image-text retrieval methods and captioner-based text-text retrieval methods. Specifically, for captioner-based methods, we first employ image captioners to generate the description for a given query image, followed by retrieving similar descriptions using the NV-Embed-v2 text embedding model [115]. For models producing panoptic captions, we ask Qwen2.5-14B to transform panoptic captions into detailed natural-language captions without bounding boxes, as NV-Embed-v2 exhibits superior performance on such descriptions. As shown in Table A1, on the challenging DOCCI dataset, our PancapChain can achieve comparable performance with the state-of-the-art MATE model [3], despite using

Table A1: Image-text retrieval results on DOCCI [4] in terms of Recall@1 and Capture metrics [41]. We report the similarity between generated and retrieved descriptions on the dimensions of object, attribute and relation using Capture. The **bold** numbers indicate the best results.

| Models | Type | R@1 | Object | Attribute | Relation |
|---|---|---|---|---|---|
| CLIP [13] | Image-Text | 16.9 | - | - | - |
| ALIGN [18] | Image-Text | 59.9 | - | - | - |
| BLIP [15] | Image-Text | 54.7 | - | - | - |
| LongCLIP [113] | Image-Text | 38.6 | - | - | - |
| MATE [3] | Image-Text | **62.9** | - | - | - |
| GIT [114] | Text-Text | 16.7 | 32.9 | 14.9 | 24.6 |
| BLIP [15] | Text-Text | 47.3 | 43.3 | 15.4 | 40.5 |
| LLaVA-1.5 [74] | Text-Text | 55.4 | 58.8 | 46.8 | 52.4 |
| ShareGPT4V [5] | Text-Text | 59.6 | 62.0 | 56.8 | 51.4 |
| PancapChain (Ours) | Text-Text | **61.9** | **65.2** | **60.3** | **53.6** |

no image-text retrieval training data or specialized module designs. Our PancapChain also outperforms state-of-the-art image captioners (*e.g.*, ShareGPT4V), demonstrating its effectiveness in capturing image details. In addition, using the Capture metrics [41], we demonstrate that PancapChain retrieves descriptions from the text corpus that are more semantically aligned with ground-truth descriptions, excelling on the dimensions of object, attribute, and relation. In summary, this experiment highlights the powerful application potential of our panoptic captioning model, as its straightforward integration with a text embedding model yields an effective image-text retrieval solution.

## B    Image Reconstruction Experiments

To qualitatively demonstrate the effectiveness of our model, we conduct an image reconstruction experiment by associating captioners with text-to-image generation models. This experiment serves as a proxy for evaluating the completeness of image descriptions, *i.e.*, if a caption captures all essential visual elements, a text-to-image model would be able to reconstruct an image similar to the original one. In this experiment, we compare our PancapChain-13B model with three baselines, namely Qwen2.5-VL-72B [86], ShareGPT4V-13B [5] and BLIP-2 [15]. Specifically, our PancapChain and Qwen2.5-VL are instructed to generate panoptic captions, ShareGPT4V generates conventional detailed captions, and BLIP-2 generates brief captions. Based on a generated caption for an input image, we adopt the text-to-image generation model PixArt-$\Sigma$ [14] to generate a new image. Figure A1 shows image reconstruction results using different captioners. As shown in the figure, PixArt-$\Sigma$ associated with our PancapChain model can generate more similar images to the original images, compared with other baseline models. For example, our model more accurately captures the person's leg in the first sample, and it more accurately identifies the dog's location in the second sample. Similarly in other examples, our model shows superiority in describing locations and attributes of primary entity instances in images, *i.e.*, the boy riding a bike in the third sample, the monk and cars in the fourth sample, and the men and horses in the fifth sample. Overall, this experiment demonstrates the effectiveness of our model in comprehensively capturing image details.

## C    More Details about PancapScore

### C.1    More Implementation Details

To evaluate models' performance in panoptic captioning, we propose a new metric named Pancap-Score, which comprehensively considers five dimensions of semantic content in panoptic captions.

| Original Image | PancapChain | Qwen2.5-VL | ShareGPT4V | BLIP-2 |
|---|---|---|---|---|

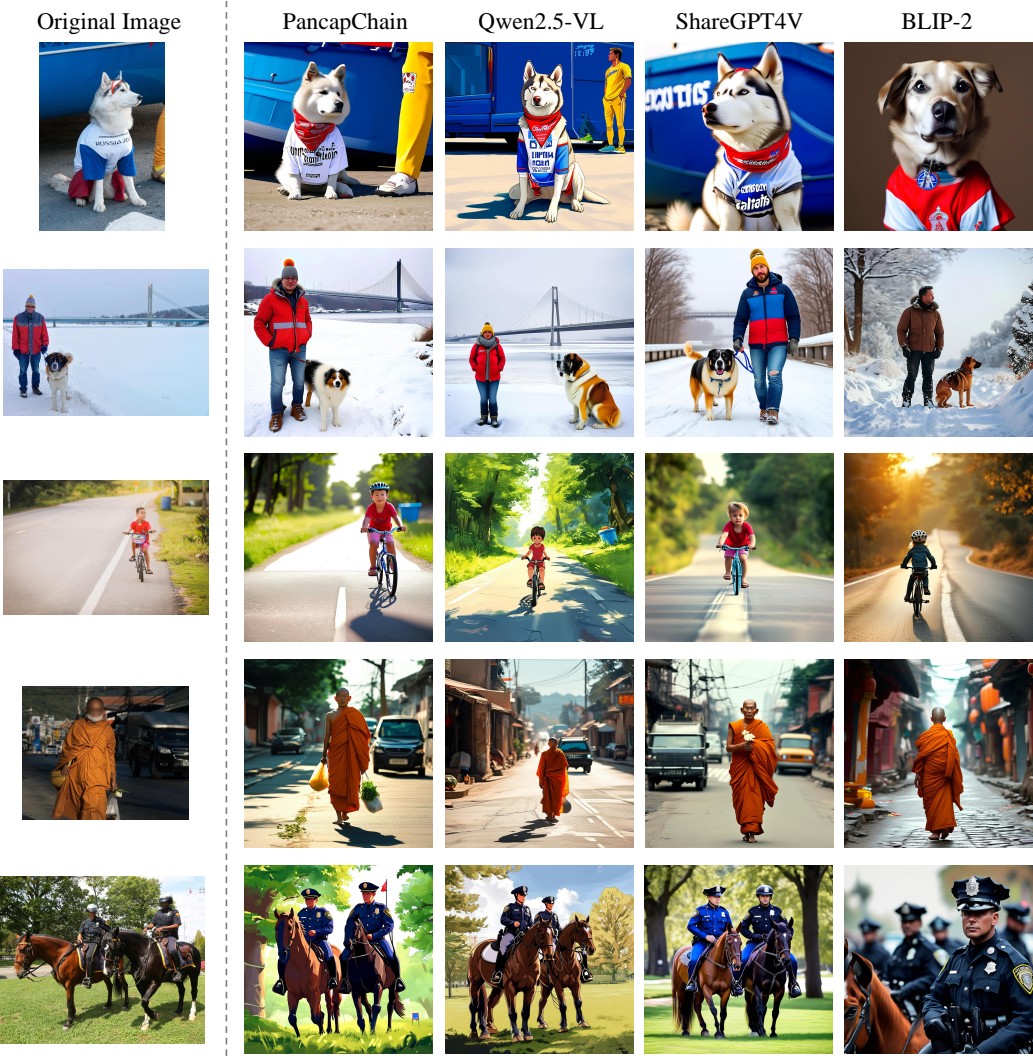

Figure A1: Image reconstruction using PixArt-Σ [14]. Compared with previous models, our PancapChain can better capture image details, and thus lead to better image reconstruction. Best viewed in color.

PancapScore consists of three steps, namely semantic content extraction, entity instance matching and instance-aware question-answering. In this part, we introduce more details about entity instance matching and instance-aware question answering.

**Entity Instance Matching.** After extracting semantic content, we conduct an entity instance matching between the generated caption and reference caption to measure the performance in semantic tagging and instance localization. To find an optimal one-to-one matching for entity instances in the two captions, we should assign a semantic similarity score to each pair of predicted entity instance and ground-truth entity instance. Formally, the semantic similarity score is given as follows:

$$s_{i,j} = \mu^2 \cdot s_{\text{eq}}(t_i, \hat{t}_j) + \mu \cdot s_{\text{sy}}(t_i, \hat{t}_j) + \cos(t_i, \hat{t}_j), \tag{A1}$$

where $\mu = 10$ is a weight coefficient to raise the priority as mentioned in our main manuscript. In Eq. (A1), $s_{\text{eq}}(t_i, \hat{t}_j)$ is a function that measures whether two tags are the same, *i.e.*, $s_{\text{eq}}(t_i, \hat{t}_j) = 1$ if $t_i$ and $\hat{t}_j$ use the same words. $s_{\text{sy}}(t_i, \hat{t}_j)$ is a function that measures whether two tags share synonymous nouns, where we use WordNet [116] to get the synonym set of semantic tags for measurement. $\cos(t_i, \hat{t}_j)$ denotes the cosine similarity between the two tags' text embeddings, where we encode the semantic tags using SentenceBERT [117]. We prioritize matching tags first by exact word matches, then by synonyms, and finally by embedding similarity, weighted by coefficient $\mu$. Based on the assigned semantic similarity scores, we can obtain the optimal one-to-one matching

between two sets of entity instances. Since this entity instance matching is a bipartite graph matching problem, we solve it using the Hungarian algorithm.

Based on the entity instance matching, we measure the model performance in semantic tagging and instance localization. First, we focus on evaluating semantic tagging based on semantic consistency. Specifically, we consider two matched instances to be semantically consistent if their semantic tags use the same words, or have synonym nouns, or have similar text embeddings, i.e., $s_{i,j} \geqslant \delta_t$. By considering all entity instances, we compute precision and recall in semantic tagging. Precision measures how many predicted instances have consistent tags according to the ground-truth instances in the reference caption, while recall measures how many ground-truth instances are accurately recognized in the generated caption. Based on precision and recall, we compute the F-score for an overall measurement for semantic tagging.

Additionally, we evaluate the performance of entity instance localization based on location consistency. Specifically, we consider two matched instances to be consistent in location, if they have consistent semantic tags and their bounding boxes have an IoU higher than a preset threshold, i.e., $s_{i,j} \geqslant \delta_t$ and $\mathrm{iou}(b_i, \hat{b}_j) \geqslant \delta_l$. Similar to semantic tagging, we compute precision and recall in instance localization. Precision measures how many predicted instances have consistent locations according to the ground-truth instances in the reference caption, while recall measures how many ground-truth instances are accurately localized in the generated caption. Finally, based on precision and recall, we obtain F-score in entity instance localization.

Overall, the proposed entity instance matching enables instance-level evaluation in semantic tagging and instance localization. It is important to highlight that some previous metrics in conventional image captioning fail to distinguish different entity instances with identical semantic tags, or they attempt to distinguish them based on entity attributes [41]. Such solutions would easily cause inaccurate assessment of semantic tagging capabilities, as there may be numerous instances of the same tag in one image. Unlike these works, our PancapScore metric distinguishes entity instances by spatial locations, which is a more straightforward and accurate way.

**Instance-aware Question Answering.** Based on entity instance matching, we evaluate models' performance in attribute, relation and global state in a question-answering manner. While entity instance matching provides a foundation for evaluating semantic tagging and localization, the question-answering approach offers a more flexible framework to assess the more nuanced dimensions of attributes, relations, and global state. First, we generate questions based on the semantic content of the generated caption, where each question verifies whether an attribute/relation/global state item is described in the reference caption. As each semantic item is associated with a unique entity instance, each instance ID in the generated caption should be mapped to the corresponding instance ID in the reference caption, as different captions assign different IDs to the same instance. For example, if the generated caption contains an item "ID 2 is red" and the "ID 2" corresponds to the "ID 3" in the reference caption, we generate a question like "Is ID 3 red?". To improve the robustness of the evaluation, we generate a pair of questions for each semantic item: one with a "Yes" answer and the other with a "No" answer.

We employ a state-of-the-art LLM to answer questions according to the reference caption. In this process, we instruct the employed LLM with a carefully designed prompt, and in-context text examples are included in the prompt to facilitate better understanding of our instructions. If the LLM outputs the same answer as the preset one, we consider the corresponding semantic item to be a correct prediction. By computing how many questions are correctly answered, we can obtain precision in predicting attributes, relations and global states. Similarly, we can measure recall on dimensions of attribute, relation and global state, where we generate questions based on the semantic content of the reference caption and instruct the employed LLM to answer questions based on the generated caption. Finally, based on precision and recall, we obtain F-score on dimensions of attribute, relation and global state.

## C.2 Human Consistency Analysis

In this part, we conduct an analysis to demonstrate the reliability of our proposed PancapScore metric. Following previous works [43, 41], we randomly sample 500 panoptic captions of test images in our SA-Pancap benchmark, which are generated by different models. We then ask human annotators to rate each caption, where ratings are given on all dimensions for each image-caption pair. Specifically,

Table A3: Statistics of the training, validation and test sets in our SA-Pancap benchmark. In the table, "Word", "Instance", "Attribute", "Relation" and "Global" denote the number of words, the number of instances, the number of attribute items, the number of relation items and the number of global state items mentioned in a caption on average. "Category" denotes the total number of entity categories in the whole set.

| | Word | Category | Instance | Attribute | Relation | Global |
|---|---|---|---|---|---|---|
| Train | 257.5 | 2429 | 6.9 | 12.7 | 8.7 | 1.5 |
| Validation | 275.9 | 729 | 6.9 | 11.9 | 8.2 | 1.4 |
| Test | 309.6 | 306 | 9.9 | 13.4 | 12.4 | 1.9 |

we ask human annotators to follow a clear and strict scoring process, designed according to our task formulation. First, we employ Qwen2.5-14B to extract semantic elements from models' panoptic captions. Then, human annotators score each caption across all five dimensions, by leveraging the ground-truth panoptic caption as reference. For each dimension in a generated caption, they carefully check every element. If a caption correctly identifies an ground-truth element in the reference, it gets a point. If it misses or gets an element wrong, it loses a point. In the end, human annotators combine the scores from all five dimensions to obtain an overall score for the caption.

After the human rating process, we compute the statistical correlation to compare the proposed PancapScore with human ratings. We use three metrics to measuring the correlation, including the Pearson correlation coefficient (PCC) $\rho$, coefficient of determination $R^2$ and Kendall's $\tau$ (Kd $\tau$). For comparisons, we select three conventional captioning metrics (*i.e.*, BELU, ROUGE and CIDEr) and one state-of-the-art open-source detailed captioning metric to construct baseline metrics, as panoptic captioning is a new task. Specifically, we first ask Qwen2.5-14B to decouple a panoptic caption into two parts, namely a detailed textual description without bounding boxes and a set of bounding boxes. Then, we evaluate the caption quality using these metrics and the localization capabilities by IoU-based score. As a result, we can obtain four baseline metrics for panoptic captioning. As shown in Table A2, these baseline metrics exhibit limited effectiveness in human consistency. The performance of Capture, which depends on a text scene graph parsing model [119] for semantic content extraction, is notably restricted in achieving human consistency and fails to fully account for all facets of semantic content, such as the global state. Conversely, our PancapScore metric obtains a substantial improvement over the baseline metrics in human consistency, underscoring its enhanced reliability.

Table A2: Correlation between panoptic captioning evaluation metrics and human judgements. The **bold** number indicates the highest human consistency among all caption metrics.

| Metrics | PCC ($\rho$) ↑ | 1-$R^2$ ↓ | Kd $\tau$ ↑ |
|---|---|---|---|
| BELU [29]+BoxScore | 0.02 | 16.03 | 0.02 |
| ROUGE [118]+BoxScore | 0.04 | 15.42 | 0.03 |
| CIDEr [31]+BoxScore | 0.01 | 16.30 | 0.02 |
| Capture [41]+BoxScore | 0.17 | 80.29 | 0.12 |
| PancapScore | **0.60** | **9.07** | **0.40** |

## D  More Details about SA-Pancap

Based on our proposed PancapEngine, we contribute a new SA-Pancap benchmark for the panoptic captioning task. We select the SA-1B dataset [101] as the data source due to its high image quality and data diversity. Overall, our SA-Pancap benchmark comprises 9,000 training and 500 validation images paired with auto-generated panoptic captions, and 130 test images paired with human-curated panoptic captions. Due to the challenging nature of panoptic captioning, we strategically select images containing fewer than 10 entity instances for the construction of SA-Pancap. Note that our PancapEngine has the capability to generate panoptic captions for images with a higher density of entity instances. As shown in Table A3, our SA-Pancap demonstrates substantial diversity in entity category, and contains rich semantic content and high-quality comprehensive panoptic captions.

For the test set, we ask human annotators to refine model-generated panoptic captions, following a rigorous curation process. Specifically, the curation process begins with the localization and identification of entity instances within a given image. Human annotators are first tasked with carefully reviewing and refining the bounding boxes and semantic tags associated with each entity instance. Following the refinement of entity instances, annotators proceed to enhance the descriptive details of each instance by refining their attribute and relation information. This involves a thorough examination and correction of the model-generated attributes (*e.g.*, color and material) and the

relationships between different instances (*e.g.*, action relation and part-whole relation). Next, the annotators proceed to refine the global state information of the image. Finally, through the systematic integration of semantic information from all five dimensions, we derive high-quality panoptic captions that accurately and comprehensively describe the visual content. According to our entity detection suite, our entity instance annotation adopts a second-level entity granularity. For instance, as shown in Figure 1 in our main manuscript, the t-shirt is treated as a single entity, and the text on the t-shirt is also recognized as a distinct entity. However, we do not further decompose the text into individual characters, avoiding recursive subdivision beyond this level.

## E  Experiment Setups

### E.1  Implementation Details

In this part, we present more details of our PancapChain model. Specifically, our model decouples the task into four stages, and constructs four types of image-prompt-text tuples for each image for training based on the ground-truth panoptic caption. We mix these four types of tuples together for training models, and we generate panoptic captions stage-by-stage during inference. The training loss is the standard auto-regressive loss (*i.e.*, next token prediction) following LLaVA [34]. The data requirements and prompts of four model stages are as follows:

**Entity Instance Localization**: Each training tuple in this part consists of an image, a prompt, and a localization text. The localization text is in the format of

"`<box>`$[x_1^1,\ y_1^1,\ x_2^1,\ y_2^1]$`</box>`, `<box>`$[x_1^2,\ y_1^2,\ x_2^2,\ y_2^2]$`</box>`,... `<box>`$[x_1^n,\ y_1^n,\ x_2^n,\ y_2^n]$`</box>`",

where $(x_1^i, y_1^i)$ and $(x_2^i, y_2^i)$ denote the top-left and bottom-right corners of the bounding box of the $i$-th ground-truth entity in the image. The prompt template for these data is "`Please localize all entities in this image`".

**Semantic Tag Assignment**: Each training tuple in this part consists of an image, a sample-specific prompt, and an instance text. The instance text is in the format of

"$t^1$ `<box>`$[x_1^1,\ y_1^1,\ x_2^1,\ y_2^1]$`</box>`, $t^2$ `<box>`$[x_1^2,\ y_1^2,\ x_2^2,\ y_2^2]$`</box>`,... $t^n$ `<box>`$[x_1^n,\ y_1^n,\ x_2^n,\ y_2^n]$`</box>`",

where $t^i$ is a text describing the ground-truth semantic tag of the $i$-th entity in the image. The prompt template for these data is "`Please specify the semantic tags of all entities based on their bounding boxes: {...}`", where "{...}" includes the localization text in the first stage, *e.g.*, "`<box>[100, 100, 200, 200]</box>, <box>[50, 100, 150, 300]</box>,... <box>[90, 75, 500, 300]</box>`".

**Extra Instance Discovery**: Each training tuple in this part consists of an image, a sample-specific prompt, and an extra instance text. During training, we randomly split the ground-truth instance set into two parts for an image. As a result, we can split the instance text into two texts, namely

"$t^1$ `<box>`$[x_1^1,\ y_1^1,\ x_2^1,\ y_2^1]$`</box>`, $t^2$ `<box>`$[x_1^2,\ y_1^2,\ x_2^2,\ y_2^2]$`</box>`,... $t^{n_1}$ `<box>`$[x_1^{n_1},\ y_1^{n_1},\ x_2^{n_1},\ y_2^{n_1}]$`</box>`"

and

"$t^1$ `<box>`$[x_1^1,\ y_1^1,\ x_2^1,\ y_2^1]$`</box>`, $t^2$ `<box>`$[x_1^2,\ y_1^2,\ x_2^2,\ y_2^2]$`</box>`,... $t^{n_2}$ `<box>`$[x_1^{n_2},\ y_1^{n_2},\ x_2^{n_2},\ y_2^{n_2}]$`</box>`"

where $n_1 + n_2 = n$. We use the first text as the extra instance text as output (for supervision), and use the second text to construct the input prompt. The prompt template for these data is "`Please specify missing entities and their locations for this image based on these specified entities: {...}`", where "{...}" is filled in with the second text.

**Panoptic Caption Generation**: Each training tuple in this part consists of an image, a sample-specific prompt, and a ground-truth panoptic caption. The ground-truth panoptic caption is from our data engine. The prompt for these data is "`Please provide a hyper-detailed description`

Table A4: Ablation study results on the validation and test sets of SA-Pancap. $S_{\text{Loc, Tag, Disc}}$ refer to our entity instance localization, semantic tag assignment and extra instance discovery stages, respectively. The caption generation stage $S_{\text{Cap}}$ is required in all cases.

| Models | Validation | | | | | | Test | | | | | |
|---|---|---|---|---|---|---|---|---|---|---|---|---|
| | Tag | Loc | Att | Rel | Glo | All | Tag | Loc | Att | Rel | Glo | All |
| Baseline | 56.38 | 29.30 | 43.06 | 31.64 | 82.47 | 168.63 | 55.12 | 30.00 | 43.09 | 28.88 | 78.91 | 164.98 |
| w/ Instance Discovery | 56.47 | 29.61 | 43.71 | 32.62 | 82.33 | 170.64 | 55.32 | 30.34 | 43.67 | 30.55 | 79.45 | 167.83 |
| w/ $S_{\{\text{Loc, Tag}\}}$ | 57.04 | 29.83 | 43.76 | 33.69 | 84.16 | 172.74 | 55.92 | 30.82 | 43.99 | 31.87 | 79.23 | 170.52 |
| Full (w/ $S_{\{\text{Loc, Tag, Disc}\}}$) | 57.56 | 30.34 | 44.78 | 34.61 | 84.59 | 175.75 | 56.45 | 31.76 | 44.46 | 32.54 | 79.85 | 173.19 |

```
for this image, including all entities, their locations, attributes, and
relationships, as well as the global image state, based on boxes and tags:
{...}
```
", where "{...}" is filled in with the complete instance text in the second stage.

Our model adopts the general LLaVA architecture [34], and it is initialized using the pre-trained ASMv2-13B [85] checkpoint, as it has good grounding capabilities. We finetune our model using LoRA (rank $r = 128$ and $\alpha = 256$), and optimize using AdamW (batch size 128, learning rate 2e-4). During inference, our model employs greedy decoding for caption generation. We train our model on the training set of our SA-Pancap for two epochs, and conduct evaluation on the validation and test sets. For our PancapScore metric, we use Qwen2.5-14B [106] as the LLM for semantic content extraction and question answering. The threshold for measuring semantic consistency is set as $\delta_t = 0.5$, the threshold for location consistency is set as $\delta_l = 0.5$, and the weight coefficient $\lambda_g$ is set as 0.1. All experiments are implemented using 4 NVIDIA RTX A6000 GPUs.

### E.2 Baseline Setups

In our comparison experiments, we select nine state-of-the-art MLLMs as baselines, including seven open-source large-scale MLLMs and two proprietary MLLMs. These models include Molmo-72B [108], LLaVA-OneVision-72B [109], Qwen2-VL-72B [107], Qwen2.5-VL-72B [86], NVLM-72B [110], InternVL-2.5-78B [111], Llama-3.2-90B [35][2], GPT-4o [78][3] and Gemini-2.0-Pro[112][4]. The cutoff date for our model comparisons is set as February 7th, 2025. We carefully design prompts for these MLLMs to unleash their panoptic captioning power, where in-context examples are introduced for better instruct-following. For a fair comparison, these prompts explicitly specify attribute, relation and global state types, which are the same as that in caption generation of our PancapEngine. In addition, we use two finetuned MLLMs as baselines for panoptic captioning, which is finetuned on our training set. We select LLaVA-1.5-13B [74] and ShareGPT4V-13B [5] here, since they have good performance in previous detailed captioning works and have the same parameter scale and architecture as our model.

## F   More Ablation Study Results

In this part, we present more results of ablation study on both validation and test sets of SA-Pancap. As shown in Table A4, model performance gets gradually better with the addition of more stages, as discussed in our main manuscript. Additionally, the findings on validation and test sets are consistent. However, additional stages do not always improve performance in the global state dimension, which relies on the overall image characteristics rather than specific entity instances. Since our proposed task decoupling approach primarily focuses on separating semantic tags, locations and other details of each entity instance, it results in modest improvements in the global state dimension. Overall, the results in the table demonstrate the effectiveness of our designs, and show that our work establishes a strong baseline for panoptic captioning to drive further advancements.

## G   Can Panoptic Captions Improve MLLMs?

To further demonstrate the effectiveness of our panoptic captions, in this part, we made an attempt to introduce panoptic captions in improving MLLMs. Specifically, we generate panoptic captions

---

[2] www.llama.com/docs/model-cards-and-prompt-formats/llama3_2/

[3] openai.com/index/gpt-4o-system-card/

[4] deepmind.google/technologies/gemini/pro/

Table A5: Results of LLaVA-1.5 by introducing panoptic captions in the pretraining stage. The results demonstrate the effectiveness of our panoptic captions in improving MLLM pretraining.

| Models | Pretraining Data (# of Samples) | Instruction Tuning Data | VizWiz | ScienceQA |
|---|---|---|---|---|
| llava-1.5-7b-lora | LLaVA-1.5 (558K) | LLaVA-1.5 | 47.8 | 68.4 |
| llava-1.5-7b-lora (Ours) | LLaVA-1.5 (464K) + Ours (94K) | LLaVA-1.5 | 49.4 | 70.2 |
| llava-1.5-7b-lora (Ours) | LLaVA-1.5 (558K) + Ours (94K) | LLaVA-1.5 | 51.2 | 70.7 |

using our PancapChain-13B model for MLLM pretraining and demonstrate its effectiveness on downstream image question answering benchmarks. First of all, we apply our model on SAM and COCO datasets to generate panoptic captions, and we obtain 94K image-caption pairs. We mix these 94K data with LLaVA-1.5's pretraining data (about 558K) and construct a dataset to pretrain LLaVA-1.5 model. Following the same pipeline as LLaVA-1.5 [74], we pretrain the model and then finetune it using LLaVA-1.5's instruction data based on LoRA. Following LLaVA-1.5, we evaluate our finetuned model on two image question answering benchmarks, *i.e.*, VizWiz [120] and ScienceQA [121], and the results are summarized in Table A5. As shown in the table, our model can obtain notable performance improvements of 3.4% and 2.3% on VizWiz and ScienceQA respectively, which demonstrates the effectiveness and superiority of our panoptic captions.

We further demonstrate that, when using the same amount of training data in pretraining as that in LLaVA-1.5, our generated panoptic captions can also benefit downstream VQA tasks. In this case, we randomly sample 464K data from LLaVA-1.5's full pretraining data (558K) and construct a new dataset consisting of 558K data (464K LLaVA-1.5's data plus 94K ours) to pretrain LLaVA-1.5 model. This new dataset has the same amount of data as that of LLaVA-1.5. As shown in Table A5, our model can obtain performance improvements of 1.6% and 1.8% on VizWiz and ScienceQA, respectively. These results demonstrate the effectiveness of our panoptic captions and confirm that our performance improvement does not merely stem from an increase in the amount of pre-training data. We believe our panoptic captions can lead to larger improvements with more data.

# H   Prompt Analysis for MLLMs

In this part, we conduct an analysis of the employed prompts for Qwen2-VL-72B [107]. Specifically, in this experiment, we use three types of prompts to instruct Qwen2-VL-72B for generating panoptic captions, as shown in Figure A2. The results of the model using different prompts are summarized in Table A6. As shown in the table, Qwen2-VL-72B obtains limited performance in solving panoptic

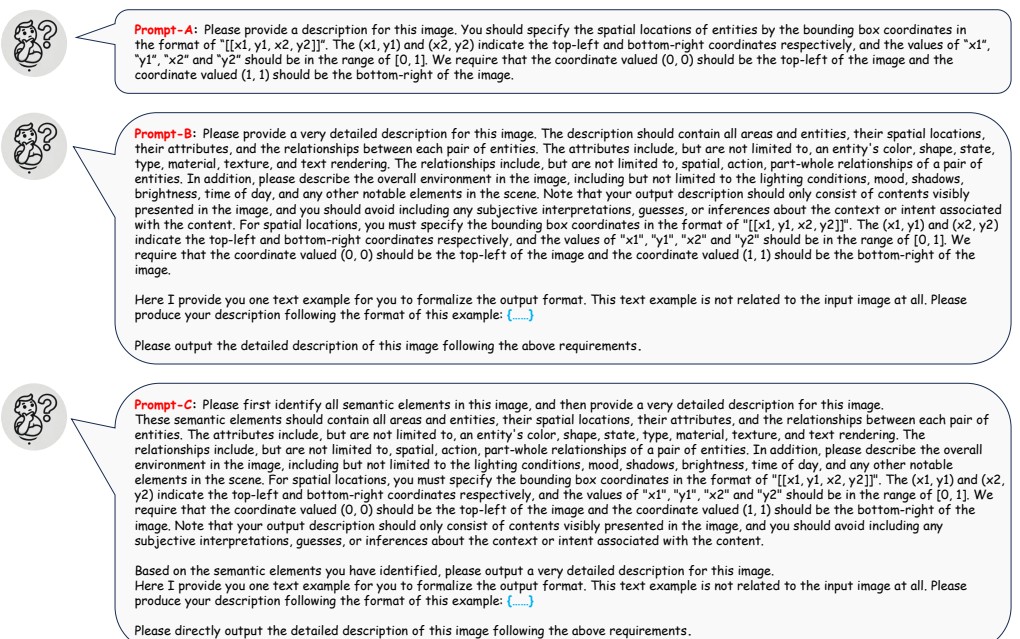

Figure A2: Three types of prompts for instructing Qwen2-VL-72B to generate panoptic captions. As shown in the figure, in-context examples are included in Prompt-B and Prompt-C.

Table A6: Results of Qwen2-VL-72B using different prompts on the validation set of SA-Pancap. In this table, Qwen-A/B/C denotes the result of Qwen2-VL-72B using Prompt-A/B/C (as shown in Figure A2). We also include the results of combining Qwen2-VL-72B with our proposed PancapChain (*i.e.*, Qwen2-VL-72B-Chain) and our PancapChain-13B model for comparison.

| Models | Tagging | Location | Attribute | Relation | Global | Overall |
|---|---|---|---|---|---|---|
| Qwen2-VL-72B-A | 48.37 | 10.25 | 34.22 | 21.29 | 82.42 | 122.37 |
| Qwen2-VL-72B-B | 50.28 | 11.56 | 36.68 | 24.66 | 86.50 | 131.82 |
| Qwen2-VL-72B-C | 49.85 | 12.92 | 37.83 | 24.71 | 86.30 | 133.96 |
| Qwen2-VL-72B-Chain | 52.17 | 15.40 | 37.90 | 25.83 | 85.62 | 139.88 |
| PancapChain-13B | 57.56 | 30.34 | 44.78 | 34.61 | 84.59 | **175.75** |

captioning, across all three cases. This demonstrates the limitations of current Multi-modal Large Language Models (MLLMs) in solving panoptic captioning tasks, as they *cannot* achieve satisfactory performance through simply prompt engineering. Additionally, we integrate Qwen2-VL-72B with our proposed PancapChain methodology by decoupling the caption generation process into multiple stages during Qwen2-VL-72B's inference. As shown in Table A6, applying our method into Qwen2-VL-72B achieves substantial improvement over the model with alternative prompts, which demonstrates the effectiveness of our idea. Our PancapChain-13B model still outperforms Qwen2-VL-72B equipped with our idea, demonstrating the superiority of our proposed model and data engine.

# I    Examples

In this part, we show some examples to demonstrate the panoptic captioning capabilities of our PancapChain model. As shown in Figure A3, our model generates panoptic captions that are largely accurate and satisfactory.

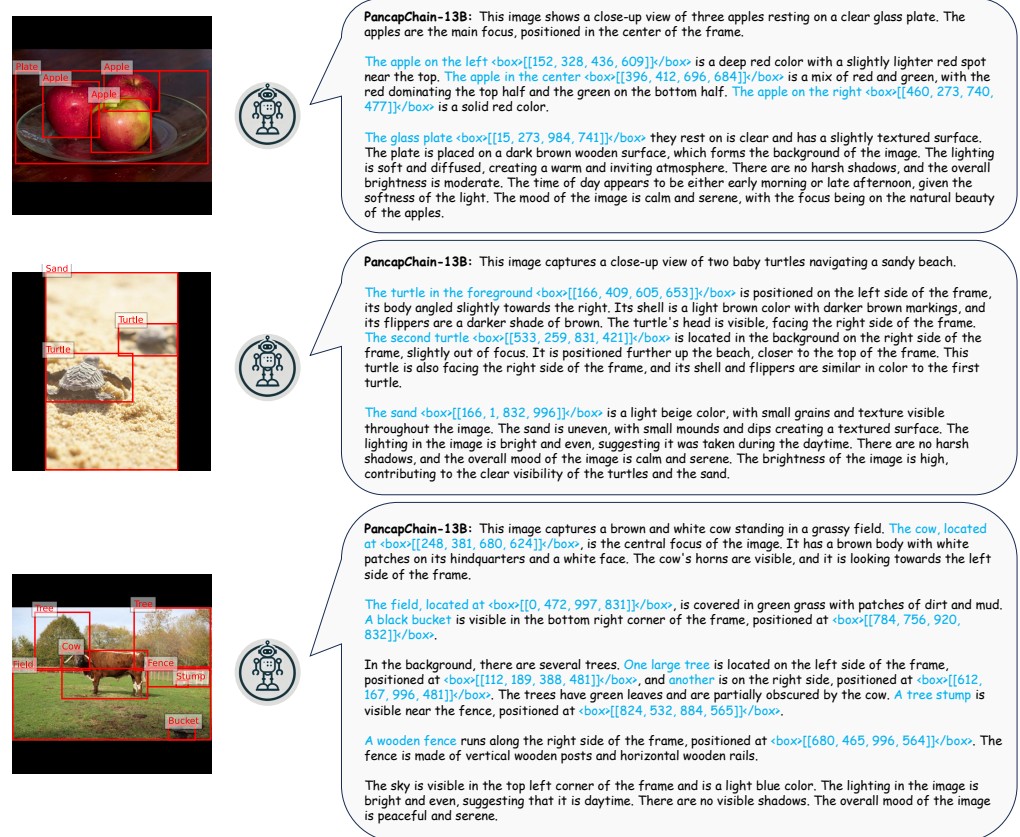

Figure A3: Example outputs of our PancapChain-13B models. For clarity, we visualize the bounding boxes and semantic tags of identified entity instances for each image.

## J   Discussions and Broader Impacts

This work introduces *panoptic captioning*, a novel task striving to seek the minimum text equivalent of images – an ambitious yet challenging goal. We take the first step towards panoptic captioning by formulating it as a task of generating a comprehensive textual description composed of five dimensions for a given image. To study this task, we contributed a evaluation metric, an effective data engine, a high-quality benchmark, and a novel decoupled learning method.

Our work reals that, despite remarkable progress in generating detailed textual descriptions for images, existing MLLMs still struggle with panoptic captioning, demonstrating their limitations in bridging image and text modalities. Also, our evaluation provides insights into the performance gap between open-source and proprietary models. By finetuning on high-quality data, our PancapChain-13B model beats state-of-the-art MLLMs, *e.g.*, InternVL-2.5-78B and Gemini-2.0-Pro, highlighting the potential of our task and model in bridging image and text modalities. Additionally, when paired with a text retriever, our model achieves superior performance in downstream image-text retrieval, demonstrating the practical utility of panoptic captioners. We believe that, developing more powerful panoptic captioners can benifit various downstream applications or learning tasks, *e.g.*, image-text alignment and multi-modal understanding.

Despite our efforts, panoptic captioning still faces numerous unresolved challenges, and our current task formulation remains an approximation, not yet fully achieving our ultimate conceptual goal of "minimum text equivalence". While our formulation is not the optimal one, it offers a straightforward and reasonable starting point. Alternative formulations for achieving minimum text equivalence, *e.g.*, using multiple bounding boxes with pose vectors to describe entity locations and states, remain underexplored and are left for future research. Nevertheless, our work is an important step towards the conceptual "minimum text equivalence" and provides a solid foundation for future development. We believe our work will act as a catalyst to accelerate the progress of developing textual representations for images.

**Broader Impacts.** First, we do not manually check every caption, so there may be some socially biased content. However, our captions are generated by advanced MLLMs designed to align with human values, significantly reducing the presence of socially biased content. Second, our work uses image data from the SA-1B dataset to establish our benchmark, inheriting certain societal impacts, such as privacy concerns. The SA-1B dataset masks most human faces in images, offering a degree of privacy protection. Users of our benchmark must adhere to the terms and conditions of the original data sources. Lastly, like many existing MLLMs, our model could be manipulated or "jailbroken" to produce unfair or inappropriate captions. This risk underscores the need to enhance MLLMs' robustness and ethical alignment, which will ensure positive impacts across diverse applications.

