# OpenReview forum: "Panoptic Captioning: An Equivalence Bridge for Image and Text"
_NeurIPS.cc/2025/Conference — NeurIPS 2025 poster_

### Official Review · Reviewer_2FRX · 2025-06-29

**Clarity:** 3
**Significance:** 2
**Originality:** 2
**Rating:** 4
**Confidence:** 3

**Summary:**

This paper introduces panoptic captioning, a challenging image captioning task that aims for minimum text equivalence of images by generating comprehensive textual descriptions. These descriptions encapsulate all entities, their locations, attributes, relationships, and global image state, emphasizing semantic entity instance-level details and location accuracy. The authors propose PancapEngine for high-quality data generation and PancapChain, a novel method that outperforms current MLLMs on the new SA-Pancap benchmark, particularly in location accuracy.

**Questions:**

1. Regarding weakness 1, please provide a more detailed discussion of the differences between your approach and methods like ImageInWords (IIW). Clarifying the specific advancements or distinct aspects of panoptic captioning beyond merely requiring bounding box information would strengthen the originality claim.

2. In Table 4's ablation study, the results for the global state (Glo) metric are not presented, and only validation set results are shown. Could you please provide the global state metrics and results on the test set for this ablation study? This additional information would offer a more comprehensive understanding of the PancapChain method's performance and contributions.

**Ethical Concerns:**

["NO or VERY MINOR ethics concerns only"]

**Final Justification:**

The authors' reply has successfully addressed my initial concerns, leading me to raise my score from borderline reject to borderline accept. Below are my initial concerns and how the authors address these.
1. Originality: My initial concern was about the similarity between the proposed data annotation pipeline and that of ImageInWords. The authors' rebuttal successfully clarifies a significant distinction: while ImageInWords relies on a human-in-the-loop annotation process, their method is fully automated. This is a major advantage and a clear contribution.
2. Method Effectiveness: The authors provided new experimental results demonstrating that their PancapChain method also shows improvements on Global metric and achieves performance gains on the test set. This evidence effectively validates the effectiveness of the proposed approach.

**Limitations:**

yes

**Quality:**

3

**Strengths And Weaknesses:**

## Strengths

1. The paper introduces panoptic captioning, a more challenging image captioning task that emphasizes semantic entity instance-level descriptions and location accuracy. I think this is helpful for advancing more precise image captioning.

2. The proposed PancapChain method demonstrates significant improvements over current MLLM models on the SA-Pancap benchmark.

## Weaknesses

1. Originality: A core aspect of the paper is its emphasis on entity-aware prompting, utilizing a two-stage captioning process involving entity detection followed by entity-aware caption generation. While the paper distinguishes itself by requiring bounding box information, similar ideas and methods have been explored in prior work, such as ImageInWords (IIW) (cited in line 158). Although IIW's final captions do not include bounding boxes, its data annotation process also involves object detection to obtain bounding box information.

2. Effectiveness of PancapChain method: While Table 3 shows PancapChain outperforming existing MLLMs on validation sets across most metrics except global state (Glo), its advantage is most significant only in location (Loc). This advantage in Loc is expected, as other MLLMs have not been trained on data explicitly including bounding box. Furthermore, this consistent advantage is somewhat diminished on the test set. Additionally, in the ablation study presented in Table 4, the numerical differences between PancapChain and the baseline appear to be modest.

---

> ### Author Rebuttal · Authors · 2025-07-31
>
> We greatly appreciate the recognition of our new task and our model's effectiveness. Below, we carefully address all the concerns and provide new perspectives to better assess our paper.
>
> Q1. Differences from ImageInWords's data annotation process [14]
>
> - Although our data annotation process shares some similarities with that of ImageInWords [14], there is an important difference between the two. ImageInWords proposes a *human-in-the-loop* framework to curate detailed image descriptions, relying on costly human labor. In contrast, our annotation process is *fully automated*, making it much cheaper and more efficient. As shown in our experiments, the superiority of our model (trained on automatically-annotated data) over state-of-the-art MLLMs demonstrates the high quality of our data.
>
> - Besides, we would like to clarify that, our contribution is not limited to a data engine for caption annotation. Overall, our work proposes a new task named panoptic captioning, serving as the initial effort to seek the minimum text equivalence of an image. Beyond our data engine, we contribute a comprehensive metric for reliable evaluation and an effective method to improve panoptic captioning. Experiments in image-text retrieval and text-to-image generation show our model's superiority over prevailing captioners (e.g., ShareGPT4V), highlighting the practical utility of our task.
>
> - Furthermore, we would like to emphasize that, introducing bounding boxes to describe locations is not trivial and offers significant value. In principle, by accurately localizing entity instances with bounding boxes, our formulation offers an effective and efficient way to describe position and occupied region of an entity instance using only several coordinate numbers. In contrast, prevailing captioning works, which vaguely specify locations by pure words, *cannot very accurately* describe instance positions. As shown in Table 1, our model outperforms the state-of-the-art detailed captioner ShareGPT4V in image-text retrieval. Also, it beats a state-of-the-art image-text alignment model ALIGN without requiring specialized training data or module designs. These results underscore the practical benefits of integrating bounding boxes in our task.
>
> - Last but not least, we would like to provide additional details about ImageInWords. ImageInWords does not open-source its models and training set, releasing only a small set of 400 images for evaluation. Based on our observation of these 400 released images, we find that these images usually features a limited number of object instances. In contrast, our dataset includes more complex scenes with a greater number of instances. As shown in Table 2, our captions are significantly longer, averaging 345.5 tokens compared to 217.2 tokens for ImageInWords, reflecting their greater detail and comprehensiveness. We will release our code, data and model.
>
> Q2. About performance improvement
>
> - First of all, we would like to clarify that, some MLLMs (e.g., Qwen-2.5-VL) involve training data explicitly including bounding box, e.g., grounding and region captioning data. In addition, we also compare our model with two baselines trained on our data. As shown in Table 3, our PancapChain outperforms LLaVA-1.5-tuned by 13.96% and ShareGPT4V-tuned by 19.05% in terms of the PancapScore metric on the validation set, which demonstrates the effectiveness of our model.
>
> - Second, we would like to clarify that our model can obtain substantial improvements *even without considering the location dimension*. The table below shows some model results in terms of an overall metric excluding the location dimension. As shown by the table, compared with state-of-the-art MLLMs, our PancapChain-13B performs much better with a significantly smaller model size, e.g., surpasses Qwen-2.5-VL-72B by 15.57% and Gemini-2.0-Pro by 11.32%.
>
> |Models (Val)|Tagging|Attribute|Relation|Global|Overall (w/o Location)|
> |:-|:-----:|:-------:|:------:|:------:|:------:|
> |Qwen-2.5-VL-72B|54.08|40.00|27.24|85.34|129.84|
> |InternVL-2.5-78B|54.68|41.81|27.41|88.37|132.74|
> |Llama-3.2-90B|52.87|39.94|27.09|83.40|128.25|
> |GPT-4o (Proprietary)|50.89|40.54|25.40|**88.85**|125.71|
> |Gemini-2.0-Pro (Proprietary)|53.79|43.14|28.52|86.50|134.09|
> |PancapChain-13B (Ours)|**57.56**|**44.78**|**34.61**|84.59|**145.41**|
> |
>
> - As shown in Table 3, our PancapChain obtains significant performance gains over state-of-the-art MLLMs on both validation and test sets. Notably, our model outperforms Qwen2.5-VL-72B (the strongest open-source MLLM here) by 26.21% on the validation set and by 18.86% on the test set. These performance gains are both substantial, as *the proprietary Gemini-2.0-Pro only outperforms Qwen2.5-VL-72B by 1.21% on the validation set and by 0.99% on the test set*. While there is some variation in performance gains between the validation and test sets, likely due to data distribution shifts, our improvement of 18.86% over Qwen2.5-VL-72B on the test set remains robust. Overall, PancapChain establishes a strong baseline for panoptic captioning and lays a solid foundation for future advancements in this field.
>
> - Finally, we would like to clarify that the performance gain of our model over our baseline is substantial. Specifically, in terms of the overall metric, our full model outperforms the baseline by 7.12% on the validation set. This improvement is notable, as *the  proprietary Gemini-2.0-Pro (which is expected to be much larger) only outperforms Qwen2.5-VL-72B by 1.21% on the validation set*. For more ablation study results, please refer to the response to the fourth comment.
>
> Q3. Differences from ImageInWords
>
> - Our work fundamentally differs from ImageInWords in both research objective and contribution. ImageInWords aims to improve detailed image captioning by a *data-centric* approach. Specifically, it carefully designs a *human-in-the-loop* framework to curate detailed image descriptions for model finetuning, relying on costly human labor. In contrast, our work proposes a new task named panoptic captioning, serving as the initial effort to seek the minimum text equivalence of an image – an ambitious yet challenging goal. We contribute a comprehensive metric for reliable evaluation, an *automated* data engine for high-quality data generation, and an effective method to improve panoptic captioning. Experiments in image-text retrieval and text-to-image generation demonstrate our model's superiority over state-of-the-art captioners (e.g., ShareGPT4V), highlighting the enormous application potential of our task. Furthermore, as shown in Table 2, our panoptic captions are significantly longer than those of ImageInWords, averaging 345.5 tokens compared to 217.2 tokens for ImageInWords, reflecting the greater detail and comprehensiveness of our panoptic captions.
>
> - Additionally, we would like to clarify that, introducing bounding boxes to describe locations is not trivial and offers significant value. In principle, by accurately localizing entity instances using bounding boxes, our formulation offers an *effective and efficient* way to describe position and occupied region of an entity instance with only several coordinate numbers. In contrast, prevailing captioning works that vaguely specify locations by pure words *cannot very accurately* describe instance positions. As shown in Table 1, our model outperforms the state-of-the-art detailed captioner ShareGPT4V in image-text retrieval. Also, it beats a state-of-the-art image-text alignment model ALIGN without requiring specialized training data or module designs. These results underscore the practical benefits of integrating bounding boxes in our task. We will add more discussions in our revision.
>
> Q4. More ablation study results
>
> Following the suggestion, we show more ablation results on both validation and test sets below, including metrics in all five dimensions. The notation aligns with that of our main manuscript. As shown in the tables, model performance gets gradually better with the addition of more stages, and the findings on validation and test sets are consistent. However, additional stages do not always improve performance in the global state dimension, which relies on the overall image characteristics rather than specific entity instances. Since our proposed task decoupling approach primarily focuses on separating semantic tags, locations and other details of each entity instance, it results in modest improvements in the global state dimension. Overall, the results in the tables demonstrate the effectiveness of our designs, and show that *our work establishes a strong baseline for panoptic captioning* to drive further advancements.
>
> |Models (Val)|Tagging|Location|Attribute|Relation|Global|Overall|
> |:-|:-----:|:------:|:-------:|:------:|:------:|:------:|
> |Baseline|56.38|29.30|43.06|31.64|82.47|168.63|
> |w/ Instance Discovery|56.47|29.61|43.71|32.62|82.33|170.64|
> |w/ $S_{\\{Loc, Tag\\}}$|57.04|29.83|43.76|33.69|84.16|172.74|
> |Full (w/ $S_{\\{Loc, Tag, Disc\\}}$)|57.56|30.34|44.78|34.61|84.59|175.75|
> |
>
> |Models (Test)|Tagging|Location|Attribute|Relation|Global|Overall|
> |:-|:-----:|:------:|:-------:|:------:|:------:|:------:|
> |Baseline|55.12|30.00|43.09|28.88|78.91|164.98|
> |w/ Instance Discovery|55.32|30.34|43.67|30.55|79.45|167.83|
> |w/ $S_{\\{Loc, Tag\\}}$|55.92|30.82|43.99|31.87|79.23|170.52|
> |Full (w/ $S_{\\{Loc, Tag, Disc\\}}$)|56.45|31.76|44.46|32.54|79.85|173.19|
> |

---

> > ### Comment · Reviewer_2FRX · 2025-08-03
> >
> > Thanks for the thorough explanations and additional experiments. Your reply has successfully addressed my concerns. I will raise my final rating to borderline accept.

---

> ### Author Response · Authors · 2025-08-03
>
> Dear Reviewer 2FRX,
>
> We are thrilled to note that your concerns have been addressed. We sincerely appreciate your dedicated time and effort in offering invaluable feedback and are encouraged by your recognition.
>
> Best

---

### Official Review · Reviewer_nxRe · 2025-07-02

**Clarity:** 3
**Significance:** 3
**Originality:** 3
**Rating:** 4
**Confidence:** 4

**Summary:**

This paper introduces a new task called panoptic captioning, which aims to generate the minimal text equivalent of an image by capturing all semantically important content. To support this task, the authors propose: (1) PancapScore, a new evaluation metric that assesses semantic content across five dimensions: semantic tags, locations, attributes, relations, and global image states. (2) PancapEngine, a data pipeline that follows a detect-then-caption strategy to build a high-quality dataset, resulting in the proposed SA-Pancap benchmark. (3) PancapChain, a multi-stage model that improves captioning quality by decoupling the generation process into step-by-step reasoning.

**Questions:**

Please refer to "weakness".

**Ethical Concerns:**

["NO or VERY MINOR ethics concerns only"]

**Final Justification:**

The authors have addressed my concerns. I would like to keep my score with borderline accept.

**Limitations:**

Please refer to "weakness".

**Quality:**

3

**Strengths And Weaknesses:**

Strengths

-	The paper introduces a novel task and a comprehensive framework including a metric (PancapScore), a benchmark (SA-Pancap), and a baseline method (PancapChain).
-	The multi-stage design of PancapChain is well-motivated and the experimental results show its effectiveness in capturing fine-grained image content.

Weaknesses

-	The discussion on potential real-world applications of this new task is relatively limited. While current image captioning models can miss key details, it is unclear whether generating descriptions of all entities is always desirable. In practice, users may prefer concise, focused summaries, and it is often possible to extract more details through follow-up queries to a MLLM. This paper should clarify the use cases where panoptic captions are preferable.
-	Some images may include background or irrelevant elements, and generating panoptic captions in such cases could reduce clarity and distract from the main content.
-	The paper mentions a four-stage process in PancapChain. However, a more detailed analysis of each stage's output would enhance transparency.

---

> ### Author Rebuttal · Authors · 2025-07-31
>
> We greatly appreciate the recognition of our novel task, comprehensive framework, well-motivated model design and model's effectiveness. We carefully address all the concerns below.
>
> Q1. Discussions on potential real-world applications
>
> Thanks for the comment. We would like to clarify that our work focuses on generating comprehensive panoptic captions for images, and our goal is independent of actual user demands, which may vary from user to user. We agree that, in some specific applications, users may prefer generating concise captions or are only interested in certain aspects of the image. However, a comprehensive description will satisfy different needs, as one can derive concise descriptions or specific user-preferred aspects from it. Meanwhile, we believe there are many real-world scenarios requiring highly comprehensive image descriptions, e.g., learning image-text alignment, cross-modal retrieval and vision navigation for the blind, which are valuable use cases for our model. Generating comprehensive panoptic captions is highly challenging, and we have demonstrated that state-of-the-art MLLMs struggle to solve this task effectively. To this end, our work aims to address this challenging panoptic captioning task. We will follow the suggestion to add more discussions about this in our revision.
>
> Q2. About background or irrelevant elements in images
>
> We would like to clarify that, background and some small elements could be valuable for certain practical applications. For instance, when a user asks MLLMs to describe the background of an image, models that generate brief captions without addressing the background cannot provide accurate responses. In addition, our model is specifically designed to generate comprehensive panoptic captions for images, rather than concise captions focusing solely on prominent elements. By generating highly detailed captions, our model enables a wide range of outputs tailored to user needs, showcasing the versatile applications of our panoptic captioning.
>
> Q3. Detailed analysis of each stage's output of PancapChain
>
> Following this suggestion, we conduct a detailed analysis to each stage's output of our PancapChain model. Specifically, we use our proposed PancapScore metric to quantitatively evaluate output quality in five dimensions. If a stage's output lacks content for a specific dimension, we mark the corresponding metric as "-". Additionally, for outputs containing only bounding boxes, we assume the semantic tagging is always accurate. The results are summarized in the table below, and we have some observations as follows:
>
> - From the second and third rows, we observe that entity localization performance decreases after incorporating semantic tagging, likely due to instances where a correct bounding box is paired with an incorrect semantic tag.
>
> - From the third and fourth rows, we find that introducing the extra instance discovery stage improves the performance of both semantic tagging and entity localization.
>
> - From the last two rows, we find that the output of caption generation stage heavily relies on the output of its preceding stage, as the semantic tagging and entity localization performance of these two stages are identical.
>
> By decoupling the challenging task into four stages, our model can obtain significant improvement over the baseline in terms of the overall metric.
>
> | Models | Tagging | Location | Attribute | Relation | Global | Overall |
> | :-| :-----: | :------: | :-------: | :------: | :------: | :------: |
> | Baseline | 56.38 | 29.30 | 43.06 | 31.64 | 82.47 | 168.63 |
> | Entity Instance Localization | - | 39.59 | - | - | - | - |
> | + Semantic Tag Assignment  | 57.04 | 29.83 | - | - | - | - |
> | + Extra Instance Discovery | 57.56 | 30.34 | - | - | - | - |
> | + Panoptic Caption Generation | 57.56 | 30.34 | 44.78 | 34.61 | 84.59 |175.75|
> |

---

> > ### Comment · Reviewer_nxRe · 2025-08-07
> >
> > Thank you for the rebuttal. The responses have addressed my concerns.

---

> ### Author Response · Authors · 2025-08-07
>
> Dear Reviewer nxRe,
>
> We are thrilled to note that your concerns have been addressed. We sincerely appreciate your dedicated time and effort in offering invaluable feedback and are encouraged by your recognition.
>
> Best

---

### Official Review · Reviewer_Hf9S · 2025-07-03

**Clarity:** 2
**Significance:** 2
**Originality:** 2
**Rating:** 4
**Confidence:** 3

**Summary:**

The paper describes a framework for Panoptic captioning of images, ie, a comprehensive text equivalent of images which includes detailed semantic descriptions of all the objects, including the attributes and location boxes in the images and relation between objects. The paper describes PanCapEngine - a method to generate high quality, PanCapChain - a model to improve captioning and PanCapScore - series of metrics to score for panoptic captioning. Experiments have been shown on the generated dataset, SA-PanCap comparing to previous methods for image captioning, and the proposed method PanCapChain shows better results.

**Questions:**

1. In Section 4, an Entity Detection Suite is mentioned for detecting entities in the images for preparing the dataset. Whereas in Section 5, no method for entity extraction is mentioned. During inference using PanCapChain, what is the method for entity extraction?

**Ethical Concerns:**

["NO or VERY MINOR ethics concerns only"]

**Final Justification:**

Rebuttal helped in clarifying doubts on differences with scene graph methods and generalizability of the method.

**Limitations:**

Yes

**Paper Formatting Concerns:**

There is no major issue with paper formatting. Some minor things:
1. Table 1 is in Introduction section, and also being referred there. There was no formal description of PanCapChain at this point. Also the table is not readable, being hidden within paragraph.
2. Table 2 and 4 similarly are nested within paragraphs and hard to read.
3. Spider graph in Figure 1 is not readable, also not sure what the graph is conveying. Can be removed or edited into a more readable version.

**Quality:**

2

**Strengths And Weaknesses:**

The paper has the following strengths:
1. The described problem of panoptic captioning is not well explored in literature. The paper has covered all the parts of the problem, including model, dataset and evaluation.
2. Experiments on SA-PanCap shows that proposed method outperforms other LLM based methods. Ablation has been shown on different parts of the method.

The weaknesses are as follows:
1. The problem of panoptic captioning looks similar to scene graph description of the image, where typically all the object and relations in the image are described. Example work:
Junhua Jia, Xiangqian Ding, Shunpeng Pang, Xiaoyan Gao, Xiaowei Xin, Ruotong Hu, Jie Nie,
Image captioning based on scene graphs: A survey, Expert Systems with Applications
The paper does not clearly mention how panoptic captioning is different than scene graph captioning.
2. Experiments are shown only on SA-PanCap dataset. Authors should show experiments on atleast one more dataset like GCG [43] to show generalizability of the method.
3. In Section 5, lines 342-345, the training loss and prompts are mentioned very briefly, without much explanation. Can authors give more details on the fine tuning step, data requirements for fine tuning and the prompts used for PanCapChain.

---

> ### Author Rebuttal · Authors · 2025-07-31
>
> We greatly appreciate the reviewer's recognition of our comprehensive coverage of the new task (including model, benchmark, and evaluation metric) and our model's effectiveness. In the responses below, we carefully address all the concerns and provide new perspectives to better assess our paper.
>
> Q1. Differences from scene-graph-based captioning [R3]
>
> Our work fundamentally differs from scene-graph-based captioning in both research objective and methodology.
>
> - According to Jia et al. [R3], scene-graph-based captioning aims to concurrently solve comprehensiveness and *controllability* issues of previous captioning methods (before 2023). Specifically, these methods first generate scene graphs and then produce captions based on structural scene graphs, thus leading to better controllability. However, their scene graph generation models are limited to recognizing predefined object and relationship categories, e.g., the widely used VisualGenome benchmark covers a limited set of 150 object categories and 50 relation categories. In contrast, our panoptic captioning requires generating a comprehensive textual description for an image, which captures all entity instances, their respective locations and attributes, relationships among entity instances, as well as global image state. Our work serves as the initial effort to seek minimum text equivalence of an image, an ambitious yet challenging goal, providing much more comprehensive understanding of the visual context than scene-graph-based captioning. To address panoptic captioning, we develop an effective data engine with a carefully designed detection suite, which leads to high-quality data covering 2400+ entity categories and a far broader range of relationships in free-from text. Please note that, *expanding the number of object and relationship categories of a benchmark is very challenging, which requires much more powerful data annotation process.*
>
> - Regarding methodology, scene-graph-based captioning methods rely on scene graph generation models built from object detection and relationship classification. These models struggle with complex scenes due to their restricted object and relationship category sets. Additionally, scene-graph-based captioning methods usually integrate various outputs to generate final captions, and may incorporate additional components (e.g., region captioning models). Thus, their performance may be limited by the caption integration process and additional models. In contrast, our panoptic captioning model is based on Multi-modal Large Language Models (MLLMs), which adopts a unified architecture and can directly output comprehensive textual descriptions without reliance on any other models. Also, we propose a simple yet effective method to improve panoptic captioning, which decouples the challenging task into multiple stages. Our experiments show that our PancapChain-13B model can beat state-of-the-art large open-source MLLMs like InternVL-2.5-78B, and even surpasses proprietary models like GPT-4o and Gemini-2.0-Pro.
>
> We will add more discussions in our revision.
>
> [R3] Image captioning based on scene graphs: A survey. Expert Systems with Applications, 2023
>
> Q2. Experiments on GCG dataset [43]
>
> Following the suggestion, we evaluate our model in grounded captioning on GCG, aiming to demonstrate the generalizability of our panoptic captioning model. For an image, the grounded captioning task in GCG requires generating a brief grounded caption (averaging fewer than 30 tokens), which primarily describes salient objects, their bounding boxes and some object attributes (e.g., color). Since our model is specifically designed for our panoptic captioning and produces longer and more comprehensive captions than those grounded captions in GCG, we assess performance by measuring the extent to which our panoptic caption covers the GCG caption for a given image. We employ a variant of our proposed PancapScore as the evaluation metric, which calculates the proportion of correctly identified ground-truth semantic items using Recall instead of the F1-score. We report metrics for semantic tagging, location, and attribute dimensions, as well as an overall metric combining all three. As shown in the table below, our PancapChain model can outperforms Qwen-2.5-VL-72B by 10.03% in the overall metric, highlighting the effectiveness and generalizability of our model.
> |Models|Tagging|Location|Attribute|Overall|
> |:-|:-----:|:------:|:-----:|:-----:|
> |Qwen-2.5-VL-72B|58.10|30.60|48.24|136.93|
> |LLaVA-1.5-13B-tuned|61.46|31.09|49.58|142.13|
> |PancapChain-13B (Ours)|**64.00**|**32.09**|**50.87**|**146.96**|
> |
>
> Q3. More details on the finetuning step, data requirements for finetuning and the prompts
>
> Sure, we present more details here. Specifically, our model decouples the task into four stages and constructs four image-prompt-text tuples for each image for training, according to the ground-truth panoptic caption. *We mix these four types of tuples together for training models (namely finetuning), and we generate panoptic captions stage-by-stage during inference.* The training loss is the standard auto-regressive loss (i.e., next token prediction) following LLaVA [31]. The data requirements and prompts of four model stages are as follows:
>
> - Entity Instance Localization: Each training tuple in this part consists of an image, a prompt, and a localization text. This localization text is in the format of "\<box\>[$x^1_1, y^1_1, x^1_2, y^1_2$]\</box\>, \<box\>[$x^2_1, y^2_1, x^2_2, y^2_2$]\</box\>,... \<box\>[$x^n_1, y^n_1, x^n_2, y^n_2$]\</box\>", where $(x^i_1, y^i_1)$ and $(x^i_2, y^i_2)$ denote the top-left and bottom-right corners of the bounding box of the $i$-th ground-truth entity in the image. The prompt for these data is "Please localize all entities in this image".
>
> - Semantic Tag Assignment: Each training tuple in this part consists of an image, a sample-specific prompt, and an instance text. This instance text is in the format of "$t^1$ \<box\>[$x^1_1, y^1_1, x^1_2, y^1_2$]\</box\>, $t^2$ \<box\>[$x^2_1, y^2_1, x^2_2, y^2_2$]\</box\>,... $t^n$ \<box\>[$x^n_1, y^n_1, x^n_2, y^n_2$]\</box\>", where $t^i$ is a text describing the ground-truth semantic tag of the $i$-th entity in the image. The prompt for these data is "Please specify the semantic tags of all entities based on their bounding boxes: {}", where "{}" includes the localization text in the first stage, e.g., "\<box\>[$x^1_1, y^1_1, x^1_2, y^1_2$]\</box\>, \<box\>[$x^2_1, y^2_1, x^2_2, y^2_2$]\</box\>,... \<box\>[$x^n_1, y^n_1, x^n_2, y^n_2$]\</box\>".
>
> - Extra Instance Discovery: Each training tuple in this part consists of an image, a sample-specific prompt, and an extra instance text. During training, we randomly split the ground-truth instance set into two parts for an image. As a result, we can split the instance text into two texts, namely "$t^1$ \<box\>[$x^1_1, y^1_1, x^1_2, y^1_2$]\</box\>, $t^2$ \<box\>[$x^2_1, y^2_1, x^2_2, y^2_2$]\</box\>,... $t^{n_1}$ \<box\>[$x^{n_1}_1, y^{n_1}_1, x^{n_1}_2, y^{n_1}_2$]\</box\>" and "$t^1$ \<box\>[$x^1_1, y^1_1, x^1_2, y^1_2$]\</box\>, $t^2$ \<box\>[$x^2_1, y^2_1, x^2_2, y^2_2$]\</box\>,... $t^{n_2}$ \<box\>[$x^{n_1}_1, y^{n_2}_1, x^{n_2}_2, y^{n_2}_2$]\</box\>" where $n_1+n_2=n$. We use the first text as the extra instance text, and use the second text to construct the prompt. The prompt for these data is "Please specify missing entities and their locations for this image based on these specified entities: {}", where "{}" is filled in with the second text.
>
> - Panoptic Caption Generation: Each training tuple in this part consists of an image, a sample-specific prompt, and a ground-truth panoptic caption. The ground-truth panoptic caption is from our data engine. The prompt for these data is "Please provide a hyper-detailed description for this image, including all entities, their locations, attributes, and relationships, as well as the global image state, based on boxes and tags: {}", where "{}" is filled in with the instance text in the second stage.
>
> We have included some implementation details in our Appendix (e.g., prompts in Section E.1), and we will add more details in our revision.
>
> Q4. Entity extraction method for PancapChain
>
> We use our PancapChain model to conduct entity extraction (without any other models), as illustrated in Section 5 Line 307-324. Specifically, our PancapChain model decouples the panoptic captioning task into four stages, and the entity extraction is accomplished by the first two stages.
>
> - In the first stage, we localize entity instances by prompting our PancapChain model to output a text in the format of "\<box\>[$x^1_1, y^1_1, x^1_2, y^1_2$]\</box\>, \<box\>[$x^2_1, y^2_1, x^2_2, y^2_2$]\</box\>,... \<box\>[$x^n_1, y^n_1, x^n_2, y^n_2$]\</box\>", where "[$x^1_1, y^1_1, x^1_2, y^1_2$]" denotes that there is an instance located in the bounding box with $(x^1_1, y^1_1)$ as its top-left corner coordinate and $(x^1_2, y^1_2)$ as its bottom-right corner coordinate.
>
> - In the second stage, we assign semantic tag to each bounding box by prompting our PancapChain model to output a text like "cat \<box\>[$x^1_1, y^1_1, x^1_2, y^1_2$]\</box\>, dog \<box\>[$x^2_1, y^2_1, x^2_2, y^2_2$]\</box\>,... bag \<box\>[$x^n_1, y^n_1, x^n_2, y^n_2$]\</box\>", where "cat [$x^1_1, y^1_1, x^1_2, y^1_2$]" denotes that there is a cat bounded by the first box.
>
> Q5. Some minor things on paper formatting
>
> Thanks for pointing out. We will fix them in our revision. In addition, we would like to clarify that, the spider figure in Figure 1 aims to show our model's superior performance in panoptic captioning. The figure shows that three state-of-the-art MLLMs (e.g., Gemini-2.0-Pro) struggle with panoptic captioning, while our proposed PancapChain performs generally better with a significantly smaller model size.

---

> ### Comment · Reviewer_Hf9S · 2025-08-08
>
> Thanks for the feedback. Regarding the comparison with Scene graph methods, the difference only seems to be in the scale of object classes, which is a valid contribution. But authors need to make it more clear in the revised paper draft, also a comparison table with previous scene graph datasets. Regarding more details on the fine tuning and entity extraction steps, the description looks a bit cumbersome and authors need to simplify it/add it in the workflow diagram. The experiment with GCG dataset helps in clarifying doubts on generalizability. Overall although the rebuttal is helping in clarifying doubts, the paper still needs a major revision. I will raise my rating to borderline accept, but authors need to incorporate the suggested edits.

---

> ### Author Response · Authors · 2025-08-09
>
> Dear Reviewer Hf9S,
>
> We are very grateful and encouraged that our responses have addressed your concerns. We are committed to reflect all the clarification, discussion and comparison in our revision.
>
> We would also like to take this opportunity to provide a bit more clarification on panoptic captioning is *far beyond the scale of object classes* when comparing with scene-graph-based captioning. First, our panoptic captioning provides a much more comprehensive description of image content, which captures *all entity instances*, their *precise locations* and *attributes*, *fine-grained relationships* among entity instances, as well as *global image state*. Second, our model is designed based on MLLMs, which provides a more unified and flexible way to address the task without reliance on any other models. Third, panoptic captioning could benefit numerous downstream applications, e.g., image-text retrieval as demonstrated in our paper. To the best of our knowledge, no existing scene-graph-based captioning methods can adequately address our proposed panoptic captioning task.
>
> Once again, we sincerely thank the reviewer for the constructive comments, which are very helpful in improving our paper.
>
> Best

---

### Official Review · Reviewer_tSAM · 2025-07-04

**Clarity:** 2
**Significance:** 2
**Originality:** 3
**Rating:** 4
**Confidence:** 4

**Summary:**

This paper introduces panoptic captioning, a new task that aims to generate a single, comprehensive textual description of an image, covering all entities, their locations, attributes, relationships, and the global scene. To support this goal, the authors propose PancapScore, a metric that evaluates captions across five semantic dimensions; PancapEngine, a data pipeline that detects diverse image entities and generates detailed captions; and SA-Pancap, a benchmark dataset with high-quality training data and a human-curated test set. They also present PancapChain, a multi-stage captioning model that sequentially processes image content and achieves state-of-the-art performance, outperforming even larger proprietary models like GPT-4o and Gemini-2.0-Pro.

**Questions:**

1. The author stresses the model’s minimal nature in the abstract and introduction, yet no experiments are presented to confirm this claim. Is there any proof demonstrating that the caption generated by model is indeed minimal and cannot be further reduced?

**Ethical Concerns:**

["NO or VERY MINOR ethics concerns only"]

**Final Justification:**

The rebuttal provides new controlled experiments showing that panoptic captions improve downstream VQA performance and clarifies the role of “minimum text equivalence” as a broader research goal. While the positioning could still be refined, the task formulation, data engine, and evaluation metric are valuable contributions, and the additional results strengthen the paper’s practical relevance. I raise my rating to Borderline Accept.

**Limitations:**

Yes.

**Quality:**

2

**Strengths And Weaknesses:**

Strengths:

1.The paper is well-written and clearly structured, with intuitive visualizations that enhance understanding.

2.The overall idea is sound, and the multi-step captioning framework is logically designed and well-motivated.

Weaknesses:

1.The experimental evaluation is somewhat limited, as it only tests on the authors' own benchmark. It would strengthen the paper to include results on standard image question answering benchmarks to assess whether the proposed method provides broader improvements.

2.The benefit of the generated captions for pretraining is not demonstrated. It would be valuable to show whether incorporating these captions into pretraining improves downstream performance.

---

> ### Author Rebuttal · Authors · 2025-07-31
>
> We greatly appreciate the reviewer's recognition of our work's sound idea, well-motivated model design and clear writing. Below, we provide careful responses w.r.t. all the concerns and provide additional perspectives to better assess our paper.
>
> Q1. Results on image question answering benchmarks
>
> Following the suggestion, we generate panoptic captions using our model for MLLM pretraining and demonstrate its effectiveness on downstream image question answering benchmarks. We would like to clarify that, our model is specifically designed for panoptic captioning, which is substantially different from VQA and thus it cannot be *directly* applied on image question answering benchmarks. Therefore, to address the concern, we conduct experiments based on pretraining. Specifically, we apply our model on SAM and COCO datasets to generate panoptic captions, and we obtain 94K image-caption pairs (considering time and computation cost, we cannot generate more in the short rebuttal phase). We mix these 94K data with LLaVA-1.5's pretraining data (about 558K) and construct a dataset to pretrain LLaVA-1.5 model. Following the same pipeline as LLaVA-1.5 [58], we pretrain the model and then finetune it using LLaVA-1.5's instruction data based on LoRA. Following LLaVA-1.5, we evaluate our finetuned model on two image question answering benchmarks, i.e., VizWiz [R1] and ScienceQA [R2], and the results are summarized in following table. As shown in the table, our model can obtain notable performance improvements of 3.4% and 2.3% on VizWiz and ScienceQA respectively, which demonstrates the effectiveness and superiority of our panoptic captions.
>
> | Models | Pretraining Data | Instruction Tuning Data | VizWiz | ScienceQA |
> | :- | :-----: | :------: | :-----: |  :-----: |
> | llava-1.5-7b-lora | LLaVA-1.5 | LLaVA-1.5 | 47.8 | 68.4 |
> | llava-1.5-7b-lora (Ours) | LLaVA-1.5 + Ours | LLaVA-1.5 |**51.2**|**70.7**|
> |
>
> The above results on image question answering benchmarks demonstrate the effectiveness of our panoptic captioning. Also, in our main manuscript, we have shown the image-text retrieval experiment to demonstrate our model's effectiveness and superiority compared with state-of-the-art image captioners (e.g., ShareGPT4V). Due to time and computation constraints in the short rebuttal phase, we can generate only 94K data for pretraining. We believe our model can lead to larger performance improvements by using more data, and we will provide these results in our revision.
>
> [R1] VizWiz Grand Challenge: Answering Visual Questions from Blind People, CVPR 2018.
>
> [R2] Learn to Explain Multimodal Reasoning via Thought Chains for Science Question Answering, NeurIPS 2022.
>
> Q2. Benefit of the generated captions for pretraining
>
> Following the suggestion, we generate panoptic captions using our panoptic captioning model for MLLM pretraining, and we find that using panoptic caption data in MLLM pretraining can benefit downstream tasks. As shown in the reply and table above (for the first comment), using panoptic captions for pretraining can improve LLaVA-1.5's performance on two downstream image question answering benchmarks, i.e., +3.4% on VizWiz and +2.3% on ScienceQA. This demonstrates the effectiveness of our model and the superiority of our panoptic captioning. Due to time and computation constraints in the short rebuttal period, we can generate only 94K data for pretraining. We believe our model can lead to larger performance improvements with more data, and we will provide these results in our revision.
>
> Q3. About the claim of minimum text equivalence
>
> We would like to clarify that, we do not claim that the caption generated by our model to have already been minimal and cannot be further reduced. As stated in Line 28-29 and Line 34-35, we claim that achieving the minimum text equivalence of an image is an *ambitious yet very challenging* goal, and our work serves as the *initial effort* towards this challenging goal.
>
> - Finding the minimum text equivalence of an image aims to develop a concise textual description that comprehensively captures its essential semantic elements. To explore this ambitious goal and make the problem tractable, our work formulates panoptic captioning as the task of generating a comprehensive textual description for an image, which captures all entity instances, their respective locations and attributes, relationships among entity instances, as well as global image state. Such a formulation serves as a reasonable *approximation* to our conceptual "minimum text equivalence", which captures basic semantic elements for comprehensiveness while excluding less critical or subtle details for conciseness, as stated in Line 35-41.
>
> - Additionally, compared with prevailing captioning works (e.g., BLIP-2’s brief captions and ShareGPT4V’s detailed captions), our panoptic captioning *mainly differs in comprehensiveness*, as stated in Line 41-47 and Line 157-161. Specifically, previous works usually vaguely specify locations by pure words, and they cannot very accurately describe the position of an object using several words. In contrast, our work accurately localizes entity instances using bounding boxes, which offers an *effective and efficient* way to describe position and occupied region of an entity instance with only several coordinate numbers (i.e., offer an efficient way to comprehensively describe positions). Experiment results in image-text retrieval (Table 1 and 5) and text-to-image generation (Figure 1 and 4) demonstrate that our panoptic captioning model outperforms BLIP and ShareGPT4V due to its comprehensiveness.

---

> ### Comment · Reviewer_tSAM · 2025-08-05
>
> Thanks for authors’ response. One question for the Lora experiment setting: Do you control the total amount of training data in pre-training stage?
>
> One more point: if the model is designed specifically for a particular task rather than for general VQA, why is minimal equivalency necessary? In my view, this minimal property should be emphasized primarily when aiming for a lossless and efficient transition between text and image modalities to enhance the overall capabilities of the MLLM. Currently, the proposed approach seems to benefit only specific tasks, such as text-image retrieval. I believe the paper could refine this claim, as the current phrasing might be somewhat misleading.

---

> ### Author Response · Authors · 2025-08-06
>
> Dear Reviewer tSAM,
>
> We sincerely appreciate the time and effort in sharing new comments and suggestions.
>
> Point 1. About the LoRA experiment setting:
>
> We conduct a new experiment by using the same amount of training data in pretraining as that in LLaVA-1.5, and we find that our generated panoptic captions can also benefit downstream VQA tasks in this case. Specifically, we randomly sample 464K data from LLaVA-1.5's pretraining data (558K) and construct a new dataset consisting of 558K data (464K LLaVA-1.5's data plus 94K ours) to pretrain LLaVA-1.5 model. This new dataset has the same amount of data as that of LLaVA-1.5. As shown in the table below, our model can obtain notable performance improvements of 1.6% and 1.8% on VizWiz and ScienceQA, respectively. This result demonstrates the effectiveness of our generated panoptic captions and confirms that our performance improvement does not merely stem from an increase in the amount of pre-training data.
> |Models|Pretraining Data (# of Samples)|Instruction Tuning Data|VizWiz|ScienceQA|
> |:-:|:-:|:-:|:-:|:-:|
> |llava-1.5-7b-lora|LLaVA-1.5 (464K)|LLaVA-1.5|47.8|68.4|
> |llava-1.5-7b-lora (Ours)|LLaVA-1.5 (464K) + Ours (94K)|LLaVA-1.5|49.4|70.2|
> |llava-1.5-7b-lora (Ours)|LLaVA-1.5 (558K) + Ours (94K)|LLaVA-1.5|51.2|70.7|
> |
>
> Point 2. Further clarification of minimum text equivalence
>
> We would like to clarify that our work specifically focuses on the panoptic captioning task, which we propose as a reasonable approximation to explore our conceptual "minimum text equivalence". This task is challenging, and we reveal that state-of-the-art MLLMs (e.g., Gemini-2.0-Pro) struggle to solve this task effectively. Additionally, our panoptic captions potentially benefit different types of downstream image-text tasks, and we have made an initial attempt to show the effectiveness of panoptic captions in improving MLLMs.
>
> - First of all, our work serves as the *initial effort* to explore an *ambitious yet challenging* goal, i.e., seeking the minimum text equivalence of images. To this end, our work formulates a new panoptic captioning task *as an approximation to this ambitious goal*, and we reveal that state-of-the-art MLLMs struggle to solve this task effectively. *To address panoptic captioning, our work specifically contributes an effective data engine, a comprehensive evaluation metric, and a simple yet effective model. We believe these contributions lay a solid foundation for future development.* We also demonstrate the application potential of panoptic captioning by taking image-text retrieval and text-to-image generation as examples, as shown in Figure 1&4 and Table 1&5.
>
> - We highlight that achieving minimum text equivalence could benefit numerous applications, e.g., general VQA, cross-modal retrieval, and vision navigation for the blind. For example, an ideally "equivalent" image description could enable an LLM to perform VQA through text-based reasoning, offering a novel VQA paradigm distinct from existing MLLMs. *Since our work is an initial step, we defer exploration of these applications to future work.* In addition, as mentioned by the reviewer, enhancing the capabilities of MLLMs could be an important application case for panoptic captioning. *We have made an initial attempt to demonstrate that our generated panoptic captions can improve MLLMs in two downstream VQA tasks* by introducing only 94K data for pretraining (please refer to the results in Point 1). We believe our panoptic captions can lead to larger improvements with more data.
>
> - Last but not least, we clarify that image-text retrieval is *the most direct application* to showcase the utility of our panoptic captioning task. Unlike traditional image-text retrieval models, we solve image-text retrieval by transforming images into texts and performing text retrieval, which represents a new way to solve multi-modal tasks. In addition, we emphasize that our image-text retrieval experiment is conducted on DOCCI [4], which consists of long description texts and images with subtle differences. This differs significantly from the classical COCO caption benchmark consisting of short captions. To address image-text retrieval on DOCCI, a captioner should produce comprehensive descriptions capturing both salient elements and fine-grained details in images. Experiment results in Table 1&5 demonstrate that our approach outperforms a state-of-the-art image-text alignment model ALIGN, without requiring specialized training data or module designs. This result is a compelling example to demonstrate the application potential of our task and model. We believe future development can benefit broader applications.
>
> We will refine the illustration of our paper to avoid misleading. Please feel free to contact us if there are any further questions or need further clarification.
>
> Best

---

### Note · Authors · 2025-08-13

Dear AC and Reviewers,

We are grateful for the reviewers' engagement in a fruitful discussion and for their recognition of our work. Here we provide a brief summary for the AC and all reviewers.

The reviewers have acknowledged the strengths of our work, including the new task (2FRX, nxRe), comprehensive framework (nxRe, Hf9S), well-motivated model design (nxRe, tSAM), model effectiveness (2FRX, nxRe, Hf9S), and clear writing (tSAM). During the rebuttal and discussion phases, the reviewers raised some points of concern, to which we have provided detailed point-by-point responses. After discussion, we have adequately addressed most concerns, and Reviewer 2FRX and Hf9S have decided to increase their rating accordingly.

Specifically, our discussions with the reviewers mainly focused on four key aspects:

- Discussions on Related Works (2FRX and Hf9S): We provided a detailed clarification highlighting the significant differences between our work and two mentioned works, which has been acknowledged by Reviewer 2FRX and Hf9S.

- Implementation and Experimental Details (2FRX, nxRe and Hf9S): We supplemented the requested experimental details and provided a thorough explanation of the implementation details of our model. These efforts have been acknowledged by Reviewer 2FRX, nxRe and Hf9S.

- Experiments on Other Tasks (Hf9S and tSAM): We conducted experiments on GCG captioning to demonstrate our model's generalizability. Additionally, we demonstrated that our panoptic captions can improve MLLMs in downstream VQA tasks by introducing our data for pretraining. These efforts have been acknowledged by Reviewer Hf9S.

- Application Potential (nxRe and tSAM): We provided a detailed clarification and presented experiments on downstream tasks to demonstrate the application potential of our task. Based on the reviewers' suggestions, we will add more discussions and refine our statements in our revision. Our clarification has been acknowledged by Reviewer nxRe.

Once again, we thank all the reviewers and the AC for the efforts in reviewing our work, which are very valuable in improving our work to benefit the wider community. We are committed to reflecting all the clarifications, discussions and comparisons in our revision based on the reviewers' suggestions.

Best regards,

Authors of Submission 6961

---

### Decision · Program_Chairs · 2025-09-17

**Decision:**

Accept (poster)

**Comment:**

The paper proposes a novel task of panoptic captioning which involves providing a comprehensive caption describing all entities, attributes and relationships on the image. The contributions of the paper also include providing an automatic data engine and evaluation protocol for this new task. The model developed by the authors achieves strong performance, surpassing that of existing multi-modal LLMs. The reviewers raised several concerns regarding the motivation of such problem setup and its differences with existing tasks like scene graph generation, the value of the generated captions for pretraining vision-language models and the improvements of the proposed model beyond the benchmark introduced in the paper. Most of the reviewers' concerns were addressed during the rebuttal and discussion period. I recommend the acceptance for this submission and encourage authors to incorporate the results from the rebuttal into the camera ready version.